# Language Reconstruction with Brain Predictive Coding from fMRI Data

## Abstract

Many recent studies have shown that the perception of speech can be decoded from brain signals and subsequently reconstructed as continuous language. However, there is a lack of neurological basis for how the semantic information embedded within brain signals can be used more effectively to guide language reconstruction. Predictive coding theory suggests the human brain naturally engages in continuously predicting future words that span multiple timescales. This implies that the decoding of brain signals could potentially be associated with a predictable future. To explore the predictive coding theory within the context of language reconstruction, this paper proposes PREDFT (**F**MRI-to-**T**ext decoding with **Pred**ictive coding). PREDFT consists of a main decoding network and a side network. The side network obtains brain predictive coding representation from related brain regions of interest (ROIs) with a self-attention module. This representation is then fused into the main decoding network for continuous language decoding. Experiments are conducted on two popular naturalistic language comprehension fMRI datasets. Results show that PREDFT achieves current state-of-the-art decoding performance on several evaluation metrics. Additional observations on the selection of ROIs, along with the length and distance parameters in predictive coding further guide the adoption of predictive coding theory for language reconstruction.

## 1 Introduction

Reconstructing natural language from functional magnetic resonance imaging (fMRI) signals offers potential insights into understanding language formation in the human brain. Recent studies have attempted to leverage brain signals with computational language models to generate coherent, naturally flowing languages Bhattasali et al. (2019); Wang et al. (2020); Affolter et al. (2020); Zou et al. (2021). This advancement is achieved by combining brain responses to linguistic stimuli with computational language models together to craft fluent language. For example, Tang et al. (2023) used a GPT (Radford et al., 2018) model to generate semantic candidates with beam search algorithm, and then brain signals are employed to select the content that is more aligned with the semantic content perceived by humans. Xi et al. (2023) proposed to obtain brain representation as the input for language model and achieves language reconstruction in a sequence-to-sequence machine translation manner (Sutskever et al., 2014).

Despite efforts in developing model architectures and utilizing language models for fMRI-to-text decoding, existing research often overlooks how natural language is encoded in the human brain and how its representation within language models. *Predictive coding* (McClelland & Rumelhart, 1981; Rao & Ballard, 1999; Friston & Kiebel, 2009) provides a powerful theory for a unified view of neural encoding and decoding. It suggests that the human brain naturally makes predictions of upcoming contents over multiple timescales when receiving current phonetic stimuli. Previous neuroscience studies (Willems et al., 2016; Okada et al., 2018) have already evidenced such speech prediction in the human brain through fMRI. Caucheteux et al. (2023) further investigated the predictive coding theory by exploring the linear mapping between language model activations and brain responses. They demonstrated that such mapping would be enhanced if predictive content is used to construct the representation in the language model. The predictive coding theory provides insights into the brain decoding process: Brain signals can potentially provide information about the upcoming content to be decoded in different time scales. However, whether the information extracted from brain predictive

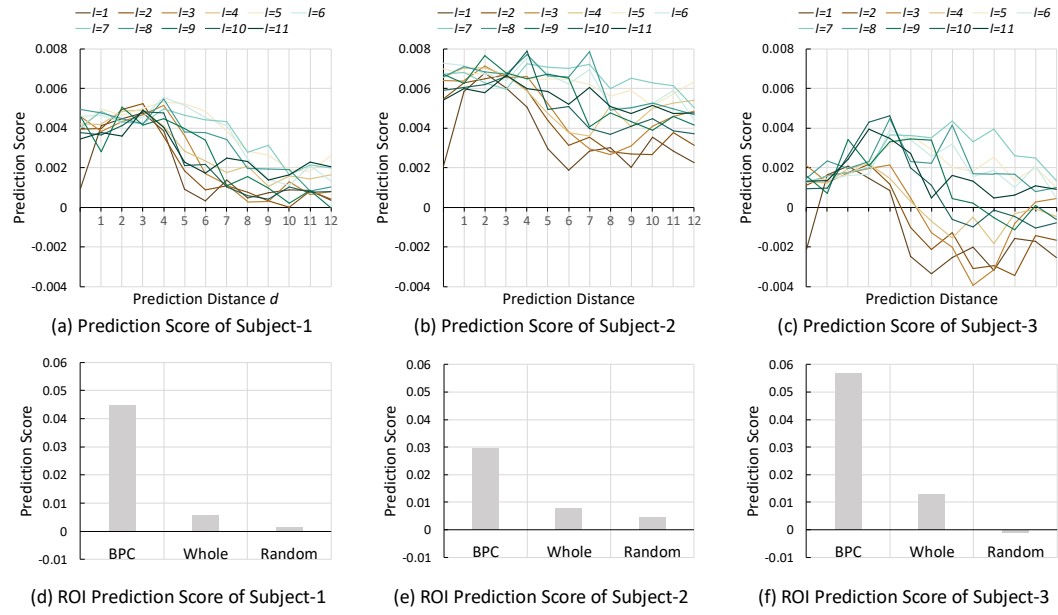

Figure 1: Results of the predictive coding verification experiment on three subjects in LeBel's dataset. For sub-figure (a) to (c), the x-axis measures the prediction distance and lines of different colors indicates prediction length. For sub-figure (a) to (c), the x-axis indicates different ROIs.

coding could help facilitate fMRI-to-text decoding and how to make use of such prediction remains an open problem.

To investigate predictive coding in fMRI-to-text decoding, we first conduct a preliminary experiment to analyze the capability of brain signals in predicting future content and their associative relationship with the representations of language models. We verify the predictive coding ability of brain signals and identify regions of interest (ROIs) in the brain that are most related to the predictive coding functions. Based on the observations, we propose PREDFT which jointly models language reconstruction and brain predictive coding. PREDFT is an end-to-end model with a main decoding network for language reconstruction and a side network for providing brain predictive coding heuristics. The main decoding network consists of an encoding model for spatial-temporal feature extraction and a Transformer (Vaswani et al., 2017) decoder for language generation. At the same time, the side network extracts and fuses ROIs related to brain predictive coding, and then builds connections to the main decoding network through attention mechanism.

Experiments are conducted on two popular naturalistic language comprehension fMRI datasets LeBel's dataset (LeBel et al., 2023) and Narratives dataset (Nastase et al., 2021). First, we present the overall decoding performance of PREDFT. We show that its decoding accuracy outperforms existing proposed methods in terms of a series of language evaluation metrics. Second, we explore whether the selection of ROIs for the side network will affect the decoding performance of PREDFT. We show that the side network brings more advancement with signals from the parietal-temporal-occipital (PTO) area, verifying its function for predictive coding. Last, we analyze the length of time for adopting brain predictive function to better understand how the human brain makes predictions over multiple timescales and its impact on language reconstruction performance.

The main contributions of this paper can be summarized as follows: (i) To the best of our knowledge, we first investigate the impact of brain predictive coding phenomenon on fMRI-to-text decoding. (ii) We propose the PREDFT model for fMRI-to-text decoding, which features effectively utilizing brain predictive coding representation to improve decoding performance through a side network and end-to-end training. (iii) Comprehensive experiments show that PREDFT benefits from the joint modeling of brain predictive coding and achieves current state-of-the-art decoding performance. Further analysis shows how brain predictive coding can be used in decoding across temporal scales and spatial brain regions.

## 2 PREDICTIVE CODING VERIFICATION

In this section, we elaborate on predictive coding in human brain by analyzing the correlation between brain responses triggered by spoken words and activations of language model with the spoken words as natural language input. Following previous study (Caucheteux et al., 2022), the brain score $R(X) = \text{corr}(f(X), Y)$ is first defined, which measures the pearson correlation between language model activations $X \in \mathbb{R}^{M \times D}$ and brain responses $Y \in \mathbb{R}^{N \times V}$. $f$ indicates a linear ridge regression model with $\ell_2$-regularization for linear mapping. $M$ and $N$ stand for the number of words and fMRI frames; $D$ and $V$ stand for the output dimension of language model and number of voxels in brain. Similar to Caucheteux et al. (2023), prediction score $P_{(d,l)}(X) = R(X \oplus X_l^d) - R(X)$ is proposed. $X_l^d$ indicates the representation of future predicted words, with **prediction length** $l$ measuring the length of continuous future words and **prediction distance** $d$ measuring the distance from current word to the first predicted future word. An example is shown in Figure 2. If the representation of current heard word "He" is denoted as $X$, then the representation of future words "and feel" is denoted as $X_2^4$. The output of pre-trained language model (Radford et al., 2019) is applied as activation and we always choose the activation of the first word within each fMRI frame as $X$. Prediction score reflects the degree of predictive coding. A positive value suggests long-range prediction helps improve the correlation between language model activations and brain responses.

Figure 2: Example of how predictive coding verification experiment is conducted.

Verification is conducted on LeBel's dataset and Narratives dataset. Following Tang et al. (2023)'s setting on LeBel's dataset, three subjects are picked for experiment. For the Narratives dataset, 230 subjects are selected. In this section, we only analyze results on the LeBel's dataset. Additional experiments on the Narratives dataset are presented in Appendix C.1. Figure 1 (a)-(c) show the prediction score of three subjects, with prediction length $l$ ranging from 1 to 11 and prediction distance $d$ ranging from 0 to 12. Figure 1 (d)-(f) show prediction score of regions of interests (ROIs). Three ROI areas are selected: "Random" indicates randomly picked ROIs. "Whole" indicates using all the ROIs from brain surface. "BPC" denotes the ROIs associated with predictive coding. Superior temporal, middle temporal, inferior parietal, supramarginal are chosen for BPC region. The specific ROIs for experiments depend on the applied cortical parcellation (Appendix A.4).

Three findings can be summarized from the experimental results. (i) For all tested prediction length $l$, the prediction score first increases and then drops when prediction distance $d$ extends. (ii) The peaking point of prediction score for too long (e.g. $l = 10, 11$) or too short (e.g. $l = 1, 2$) prediction lengths comes earlier when prediction distance $d$ extends. A proper prediction length, such as $l = 4, 5, 6$, typically results in a higher prediction score compared to excessively long or short prediction lengths. (iii) Prediction score of ROIs related to predictive coding is significantly higher than that of the entire brain or randomly selected ROIs.

The above verification experiment highlights the correlation between representation of language model and brain predictive coding. One possible explanation for the phenomenon is that both language models and brain predictive coding are similar in the objective of upcoming words prediction. This inspires us with the following motivation: While predictive coding has been verified from the perspective of brain and language model alignment, could it help in reconstructing

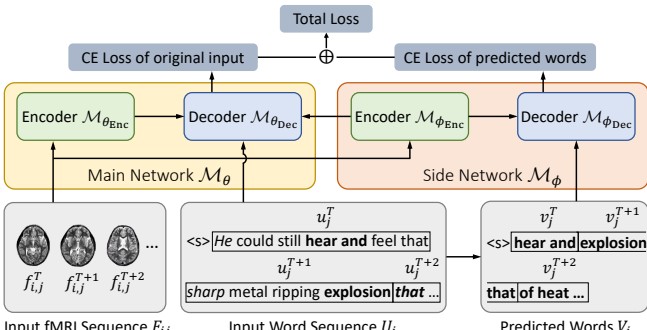

Figure 3: The framework of PREDFT in the training stage.

natural language from brain signals? We propose PREDFT for investigating the effectiveness of utilizing brain prediction in fMRI-to-text decoding. Details are introduced in the next section.

# 3 METHODOLOGY

We first formalize
the fMRI-to-text de-
coding task. Given
a naturalistic lan-
guage comprehen-
sion fMRI dataset
$\mathcal{D} := \{\langle F_{i,j}, U_j \rangle\}$,

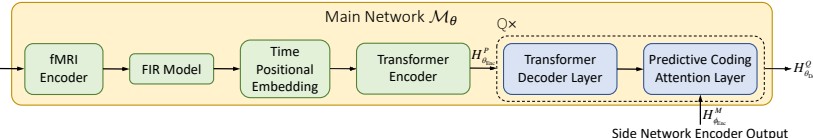

Figure 4: The main decoding network of PREDFT.

where $U_j$ is the $j$-th period of text stimuli and $F_{i,j}$ is the fMRI images collected while the $i$-th subject is hearing the text stimuli $U_j$. The fMRI-to-text decoding task aims to build a model $\mathcal{M}$ that decodes $U'_j = \mathcal{M}(F_{i,j})$ to maximize the text similarity between $U'_j$ and $U_j$. Specifically, the text stimuli $U_j := \{u_j^T, u_j^{T+1}, \ldots, u_j^{T+k}\}$ contains $k+1$ text segments of auditory content presented to test subject from time step $T$. Similarly, $F_{i,j} := \{f_{i,j}^T, f_{i,j}^{T+1}, \ldots, f_{i,j}^{T+k}\}$ consists of the same number of continuous fMRI images, and each $f_{i,j}^t$ matches $u_j^t$ at corresponding time step $t$. An example is shown in the input side of Figure 3. We propose PREDFT, which integrates brain predictive coding in the language reconstruction process. As shown in Figure 3, PREDFT is denoted as $\mathcal{M}_{\theta,\phi}$, containing a main network $\mathcal{M}_\theta$ for decoding and a side network $\mathcal{M}_\phi$ for predictive coding. We first introduce the main decoding network $\mathcal{M}_\theta$ which can reconstruct accurate text from fMRI with computational language models. Then we elaborate on the side network $\mathcal{M}_\phi$ which extracts and exploits brain predictive coding. Finally, the fusion of brain predictive coding representation and the joint training of $\mathcal{M}_\theta$ and $\mathcal{M}_\phi$ are detailed. A notation table is displayed in Table 3.

## 3.1 MAIN NETWORK FOR DECODING

As shown in Figure 3 and 4, the main network $\mathcal{M}_\theta$ consists of an encoder $\mathcal{M}_{\theta_{\text{Enc}}}$ and a decoder $\mathcal{M}_{\theta_{\text{Dec}}}$. The encoder $\mathcal{M}_{\theta_{\text{Enc}}}$ is stacked with fMRI encoder, finite impulse response (FIR) model (Huth et al., 2016), and Transformer encoder (Vaswani et al., 2017). As shown in Figure 5, the fMRI encoder is designed differently for two types of fMRI image. 4D volumetric fMRI image $F_{i,j} \in \mathbb{R}^{w \times h \times d \times (k+1)}$ where $w, h, d, k+1$ represents the width, height, depth and time steps records the activity of the whole brain. Voxel-level normalization is first applied for each image $f_{i,j}^t \in F_{i,j}$ (detailed in Appendix A),

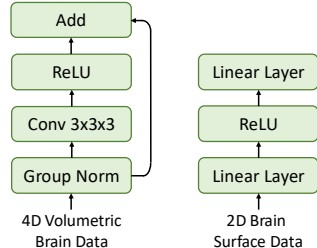

Figure 5: The fMRI encoder with different types of fMRI images as input.

and the fMRI image after normalization is denoted as $\hat{f}_{i,j}^t$, which is then fed into the 3D-CNN module. The 3D-CNN module contains $L$ layers of group normalization (Wu & He, 2018), ReLU activation, and convolution layer (LeCun & Bengio), with residual connection (He et al., 2016). The size of fMRI image $\hat{f}_{i,j}^t$ is progressively reduced by convolution layer and finally downsized to $\hat{f}_{i,j}^t \in \mathbb{R}^{w' \times h' \times d' \times c}$ where $c$ is the number of output channels. A flatten layer and a linear layer are used to obtain a one-dimensional vector $x_{i,j}^t \in \mathbb{R}^{d_m}$ as the output of the 3D-CNN module. 2D fMRI image $F_{i,j} \in \mathbb{R}^{d_s \times (k+1)}$ records the activity of brain surface. For this situation, we directly apply linear layers to gradually reduce the dimension of each image $f_{i,j}^t \in F_{i,j}$. The output $x_{i,j}^t \in \mathbb{R}^{d_m}$ remains the same dimension as the output of 3D-CNN module.

After the fMRI encoder, FIR model $g_t$ is applied to compensate for the latency of blood-oxygen-level-dependent (BOLD) signal. For $k+1$ continuous fMRI images with $t \in \{T, T+1, \ldots, T+k\}$, the temporal transformation $g_t$ concatenates $k-k^*$ future fMRI images to form the representation at time step $t$:

$$g_t : \mathbb{R}^{k \times d_m} \to \mathbb{R}^{k^* \times (d_m(k-k^*))}$$
$$x_{i,j}^t \mapsto \text{concat}(x_{i,j}^t, x_{i,j}^{t+1}, \ldots, x_{i,j}^{t+(k-k^*)}), t \in \{T, T+1, \ldots, T+k^*\}. \tag{1}$$

where $k-k^*$ is the number of delays. A linear layer $W \in \mathbb{R}^{(d_m(k-k^*)) \times d_m}$ is used to fuse delayed brain responses and recover $x_{i,j}^t$ back to its original dimension $d_m$. After learning the spatial features of fMRI images, the representations with learnable time positional embeddings, denoted as $H_{\theta_{\text{Enc}}}^0 \in \mathbb{R}^{k^* \times d_m}$, are sent into a Transformer encoder to capture temporal features within given

intervals. The output of the Transformer encoder is $H^P_{\theta_{\text{Enc}}} = \mathcal{M}_{\theta_{\text{Enc}}}(H^0_{\theta_{\text{Enc}}})$, where $P$ is the number of Transformer encoder layers.

Finally, the output from $\mathcal{M}_{\theta_{\text{Enc}}}$, i.e., $H^P_{\theta_{\text{Enc}}}$, is fed into the decoder $\mathcal{M}_{\theta_{\text{Dec}}}$. $\mathcal{M}_{\theta_{\text{Dec}}}$ contains modules within a standard Transformer decoder consisting of masked self-attention layers and encoder-

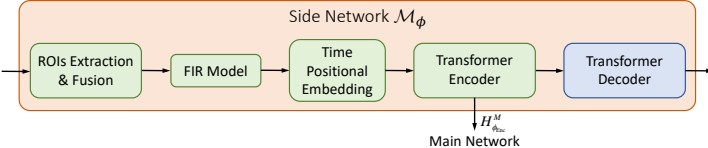

Figure 6: The side network of PREDFT.

decoder attention layers. Besides, additional predictive coding attention layers are designed to integrate brain predictive coding representations inherited from the side network for improving decoding accuracy. More details about the predictive coding attention layer will be introduced in Section 3.3. The input word sequence $U_j$ is tokenized and sent into a word embedding layer to obtain representations $H^0_{\theta_{\text{Dec}}}$. We denote the input of the $(l+1)$-th self-attention layer as $H^l_{\theta_{\text{Dec}}}$, so the self-attention is calculated by

$$\text{Self-Attn}(H^l_{\theta_{\text{Dec}}}W^l_Q, H^l_{\theta_{\text{Dec}}}W^l_K, H^l_{\theta_{\text{Dec}}}W^l_V) = \text{softmax}(\frac{H^l_{\theta_{\text{Dec}}}W^l_Q(H^l_{\theta_{\text{Dec}}}W^l_K)^\top}{\sqrt{d_k}})H^l_{\theta_{\text{Dec}}}W^l_V, \quad (2)$$

where $W^l_Q, W^l_K, W^l_V \in \mathbb{R}^{d_m \times d_k}$ are the parameter matrices of projecting query, key, value in the $(l+1)$-th corresponding layer (i.e., here is the self-attention layer) for simplicity. The encoder-decoder attention aims to integrate fMRI representations. It takes $H^l_{\theta_{\text{Dec}}}$ as query and $H^P_{\theta_{\text{Enc}}}$ for key and value:

$$\text{ED-Attn}(H^l_{\theta_{\text{Dec}}}W^l_Q, H^P_{\theta_{\text{Enc}}}W^l_K, H^P_{\theta_{\text{Enc}}}W^l_V) = \text{softmax}(\frac{H^l_{\theta_{\text{Dec}}}W^l_Q(H^P_{\theta_{\text{Enc}}}W^l_K)^\top}{\sqrt{d_k}})H^P_{\theta_{\text{Enc}}}W^l_V. \quad (3)$$

The design of masks for self-attention and encoder-decoder attention remains the same as vanilla Transformer. The output of $\mathcal{M}_{\theta_{\text{Dec}}}$ is denoted as $H^Q_{\theta_{\text{Dec}}}$ where $Q$ is the number of decoder layers.

## 3.2 SIDE NETWORK FOR PREDICTIVE CODING

The idea of designing a side network $\mathcal{M}_\phi$ for representing brain predictive coding is motivated by predictive coding theory (McClelland & Rumelhart, 1981; Rao & Ballard, 1999; Friston & Kiebel, 2009), which indicates the human brain naturally makes predictions about future words over multiple timescales. Since brain predictive coding has been verified from the perspective of brain and language model alignment (the linear mapping between language model activations and brain responses), we seek to exploit it by training a neural network to well represent regions involved in prediction, and fusing brain predictive coding representations in fMRI-to-text decoding.

The side network $\mathcal{M}_\phi$ consists of an encoder $\mathcal{M}_{\phi_{\text{Enc}}}$ to represent regions of interests (ROIs) related to predictive coding, and a decoder $\mathcal{M}_{\phi_{\text{Dec}}}$ to learn mapping between ROIs representations and predicted words. As shown in Figure 3 and 6, the encoder $\mathcal{M}_{\phi_{\text{Enc}}}$ takes $F_{i,j} := \{f^T_{i,j}, f^{T+1}_{i,j}, \ldots, f^{T+k}_{i,j}\}$ as input. The ROIs extraction layer generates $R_{ij} := \{r^T_{i,j}, r^{T+1}_{i,j}, \ldots, r^{T+k}_{i,j}\}$ from $F_{i,j}$. Each $r^t_{i,j} \in \mathbb{R}^{d_r}$ extracted from $f^t_{i,j}$ is the concatenation of ROIs related to brain predictive coding as verified in Section 2. The ROIs fusion layer is a fully connected feed-forward network that outputs the representation $r^t_{i,j} \in \mathbb{R}^{d_m}$. The same FIR model in the main decoding network is applied to compensate for the delays of the BOLD signal. Learnable time positional embedding is added to $r^t_{i,j}$ before entering into the Transformer encoder. The output of Transformer encoder is denoted as $H^M_{\phi_{\text{Enc}}}$, serving as the representation of brain predictive coding. $H^M_{\phi_{\text{Enc}}}$ plays an essential role in PREDFT, as it will be fused into the main network to verify the effectiveness of brain predictive coding in the fMRI-to-text decoding task.

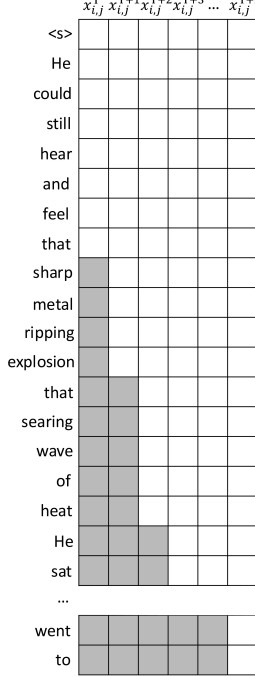

Figure 7: Mask of predictive coding attention.

The side network decoder $\mathcal{M}_{\phi_{\text{Dec}}}$ consists of Transformer decoder layers. It takes predicted future words $V_j := \{v_j^T, v_j^{T+1}, \ldots, v_j^{T+k}\}$ as input. Each $v_j^t$ is extracted from original input word sequence $u_j^t$, and stands for $l$ future words with prediction distance $d$ (recall the definition in Section 2). The side network decoder $\mathcal{M}_{\phi_{\text{Dec}}}$ follows the conventional practice of masked self-attention and encoder-decoder attention, which has been elaborated in Equation 2 and Equation 3 of Section 3.1.

### 3.3 PREDICTION FUSION AND JOINT TRAINING

This subsection details how the brain predictive coding representation $H_{\phi_{\text{Enc}}}^M$ from the side network is fused into the main decoding network. As shown in Figure 4 and 6, the brain predictive coding representation $H_{\phi_{\text{Enc}}}^M$, which is the output of $\mathcal{M}_{\phi_{\text{Enc}}}$, plays as key and value for the predictive coding attention module in the main network. The query of predictive coding attention layer is the output from the previous Transformer decoder layer. The predictive coding attention is formularized as:

$$\text{PC-Attn}(H_{\theta_{\text{Dec}}}^l W_Q^l, H_{\phi_{\text{Enc}}}^M W_K^l, H_{\phi_{\text{Enc}}}^M W_V^l) = \text{softmax}(\frac{H_{\theta_{\text{Dec}}}^l W_Q^l (H_{\phi_{\text{Enc}}}^M W_K^l)^\top}{\sqrt{d_k}}) H_{\phi_{\text{Enc}}}^M W_V^l. \quad (4)$$

The mask $\mathbf{M}_{\text{pc}} \in \mathbb{R}^{k_t \times k^*}$ of predictive coding attention is shown in Figure 7. $k_t$ and $k^*$ are the numbers of input tokens and fMRI signals, respectively. The predictive coding attention mask $\mathbf{M}_{\text{pc}}$ is designed in this way: For each token in the text fragment $u_j^t$, all the predictive coding representations after time step $t$ are allowed to attend, while previous representations are masked.

As shown in Figure 3, PREDFT is trained in an end-to-end manner. The main decoding network $\mathcal{M}_\theta$ and the side network $\mathcal{M}_\phi$ share the same word embedding layer, whose parameters are only updated with the gradient flow from $\mathcal{M}_\theta$ during training. The training objective follows a left-to-right auto-regressive language modeling manner for both $\mathcal{M}_\theta$ and $\mathcal{M}_\phi$. Following $\mathcal{M}_\theta$ and $\mathcal{M}_\phi$ are two language model heads. The cross-entropy training loss for $\mathcal{M}_\theta$ is

$$\mathcal{L}_{\text{Main}} = -\sum_{t=1}^n \log P(y_t|y_{<t}, U_j; \theta), \quad (5)$$

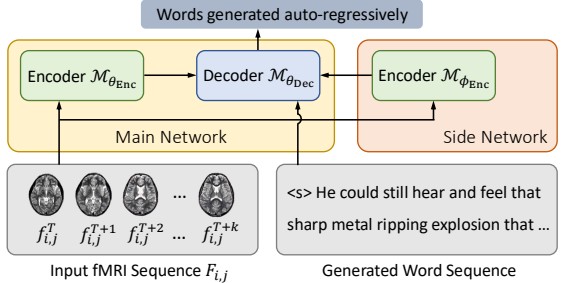

Figure 8: The framework of PREDFT in the inference stage. The decoder in side network is discarded.

where $U_j$ is the input and $y_t$ is the $t$-th generated token. Similarly, the training loss for $\mathcal{M}_\phi$ is

$$\mathcal{L}_{\text{Side}} = -\sum_{t=1}^n \log P(z_t|z_{<t}, V_j; \phi), \quad (6)$$

where $V_j$ is the input of side network and $z_t$ is the $t$-th generated token. The joint training of $\mathcal{M}_\theta$ and $\mathcal{M}_\phi$ is to optimize the total loss $\mathcal{L} = \mathcal{L}_{\text{Main}} + \lambda \mathcal{L}_{\text{Side}}$, where $\lambda$ is a hyper-parameter.

During the inference stage, the decoder in side network is discarded. The purpose of this decoder is to assist the training of encoder for obtaining predictive coding representation. Once the encoder has been trained, the decoder is no longer necessary. As illustrated in Figure 8, the input is fMRI sequence $F_{i,j}$, and the decoder in main network is responsible for generating words in an auto-regressive manner, incorporating fMRI representation and predictive coding representation.

## 4 EXPERIMENTAL SETTINGS AND RESULTS

We conduct extensive experiments to (i) evaluate the decoding performance of PREDFT (ii) analyze how brain predictive coding improves PREDFT. First, we introduce the experimental setups, including baselines and evaluation metrics in Section 4.1. The selection of hyper-parameters and more details are detailed in Appendix A. Then we present the decoding performance, regions of interest selection analysis, and prediction length and distance analysis in Section 4.2. We also elaborate more experimental analyses including decoding error distribution in Appendix D, ablation study in Appendix E, and case analysis in Appendix F.

Table 1: The performance of different models in within-subject fMRI-to-text decoding in LeBel's dataset. 10 continuous fMRI images (equals to 20 seconds) are sampled for decoding.

| | Models | BLEU-1 | BLEU-2 | BLEU-3 | BLEU-4 | ROUGE1-R | ROUGE1-P | ROUGE1-F | BERTScore |
|---|---|---|---|---|---|---|---|---|---|
| **Sub-1** | Tang's (Tang et al., 2023) | 22.25 | 6.03 | 0.83 | 0.00 | 20.16 | 19.12 | 19.44 | 80.84 |
| | BrainLLM (Ye et al., 2023) | 24.18 | 8.36 | 3.06 | 1.11 | 24.17 | 19.31 | 21.16 | **83.26** |
| | MapGuide (Zhao et al., 2024) | 27.11 | 10.02 | 3.78 | 1.54 | **25.17** | 24.64 | 24.83 | 82.66 |
| | PREDFT w/o SideNet | 27.91 | 10.26 | 3.50 | 1.29 | 18.59 | 49.00 | 26.82 | 81.35 |
| | PREDFT | **34.95** | **14.53** | **5.62** | **1.78** | 23.79 | **49.95** | **32.03** | 82.92 |
| **Sub-2** | Tang's (Tang et al., 2023) | 23.05 | 6.65 | 1.83 | 0.00 | 20.85 | 19.54 | 20.01 | 81.33 |
| | BrainLLM (Ye et al., 2023) | 23.69 | 8.06 | 2.37 | 0.00 | 23.63 | 19.29 | 21.02 | **83.40** |
| | MapGuide (Zhao et al., 2024) | 26.40 | 9.68 | 2.78 | 0.97 | 26.72 | 21.13 | 23.65 | 82.78 |
| | PREDFT w/o SideNet | 26.23 | 9.54 | 3.46 | **1.44** | **50.28** | 17.41 | 25.69 | 81.42 |
| | PREDFT | **32.46** | **11.77** | **3.95** | 0.84 | 24.90 | **38.43** | **30.01** | 82.52 |
| **Sub-3** | Tang's (Tang et al., 2023) | 23.08 | 6.83 | 2.41 | 0.82 | 21.66 | 20.07 | 20.66 | 81.50 |
| | BrainLLM (Ye et al., 2023) | 24.90 | 10.15 | **4.76** | 1.75 | 24.15 | 19.49 | 21.34 | **83.82** |
| | MapGuide (Zhao et al., 2024) | 26.41 | 9.97 | 3.71 | 1.25 | **25.33** | 23.91 | 24.53 | 82.84 |
| | PREDFT w/o SideNet | 26.89 | 10.11 | 3.84 | **1.78** | 15.72 | **55.13** | 24.31 | 81.48 |
| | PREDFT | **33.22** | **12.91** | 4.29 | 1.76 | 23.22 | 44.31 | **30.24** | 82.11 |

Table 2: The performance of different models in cross-subject fMRI-to-text decoding in Narratives dataset. Length denotes the length of time windows for continuous fMRI frames.

| Length | Models | BLEU-1 | BLEU-2 | BLEU-3 | BLEU-4 | ROUGE1-R | ROUGE1-P | ROUGE1-F | BERTScore |
|---|---|---|---|---|---|---|---|---|---|
| 10 | UniCoRN (Xi et al., 2023) | 20.64 | 5.03 | 1.40 | 0.45 | **15.56** | 25.47 | 19.23 | 75.35 |
| | PREDFT w/o SideNet | 18.08 | 3.98 | 1.05 | 0.28 | 14.96 | 26.21 | 18.96 | 75.26 |
| | PREDFT | **24.73** | **8.39** | **3.92** | **1.86** | 14.07 | **35.28** | **19.53** | **78.52** |
| 20 | UniCoRN (Xi et al., 2023) | 18.02 | 4.71 | 1.32 | **0.4** | 18.01 | **29.46** | 20.82 | 74.88 |
| | PREDFT w/o SideNet | 20.37 | 3.86 | 1.03 | 0.19 | 17.42 | 22.15 | 19.45 | 75.16 |
| | PREDFT | **25.98** | **5.61** | **1.36** | 0.21 | **19.61** | 25.43 | **22.09** | **78.20** |
| 40 | UniCoRN (Xi et al., 2023) | 21.76 | 5.43 | 1.17 | 0.34 | **19.76** | 35.33 | 25.30 | 74.40 |
| | PREDFT w/o SideNet | 18.01 | 4.72 | 1.27 | 0.34 | 16.41 | 34.36 | 22.16 | 75.07 |
| | PREDFT | **27.80** | **8.29** | **2.00** | **0.54** | 19.53 | **38.95** | **25.96** | **78.63** |

## 4.1 BASELINES AND EVALUATION METRICS

We test both within-subject and cross-subject fMRI-to-text decoding tasks in experiment (detailed in Appendix A.2). Following the setting in Tang et al. (2023), the LeBel's dataset (LeBel et al., 2023) is used for within-subject decoding. Tang's model (Tang et al., 2023), BrainLLM (Ye et al., 2023), and MapGuide (Zhao et al., 2024) are selected as compared methods. The Narratives dataset (Nastase et al., 2021) contains more subjects and is usually used for cross-subject decoding. UniCoRN (Xi et al., 2023) is selected as baseline. Detailed introduction of the compared methods are presented in Appendix A.1. For experiments on LeBel's dataset, we show results of different subjects, while for experiments on Narratives dataset, we show results of different fMRI sequence lengths.

Automatic evaluation metrics including BLEU (Papineni et al., 2002), ROUGE (Lin, 2004) and BERTScore (Zhang et al., 2019) are applied to measure the decoding performance of different models. BLEU measures the n-gram overlap between decoded content and ground truth. ROUGE-N comparing the consistency of N-grams between the decoded content and the ground truth. BERTScore measures semantic similarity between decoded content and ground truth through a BERT model. Specifically, BERTScore-F1, BLEU at different cutoffs of 1/2/3/4 and the precision, recall, and F1-score of ROUGE-1 are adopted in our experiments. More details are shown in Appendix A.3.

## 4.2 DECODING PERFORMANCE

We compare the decoding performance of different models on automatic evaluation metrics. The results of within-subject decoding in LeBel's dataset are shown in Table 1. Ten continuous fMRI images, corresponding to a 20s time interval with a repetition time (TR) of 2s, are sampled for experiments. PREDFT outperforms all the compared models on three tested subjects in BLEU-1 and ROUGE1-F, and achieves a maximum 34.95% BLEU-1 score and 32.03% ROUGE1-F score. As to BERTScore, PREDFT maintains a very narrow gap to the best performed model. We surprisingly find the PREDFT without side network also beats some baseline models. And PREDFT significantly benefits from the incorporation of side network, demonstrating the feasibility of applying predictive coding to improve decoding accuracy.

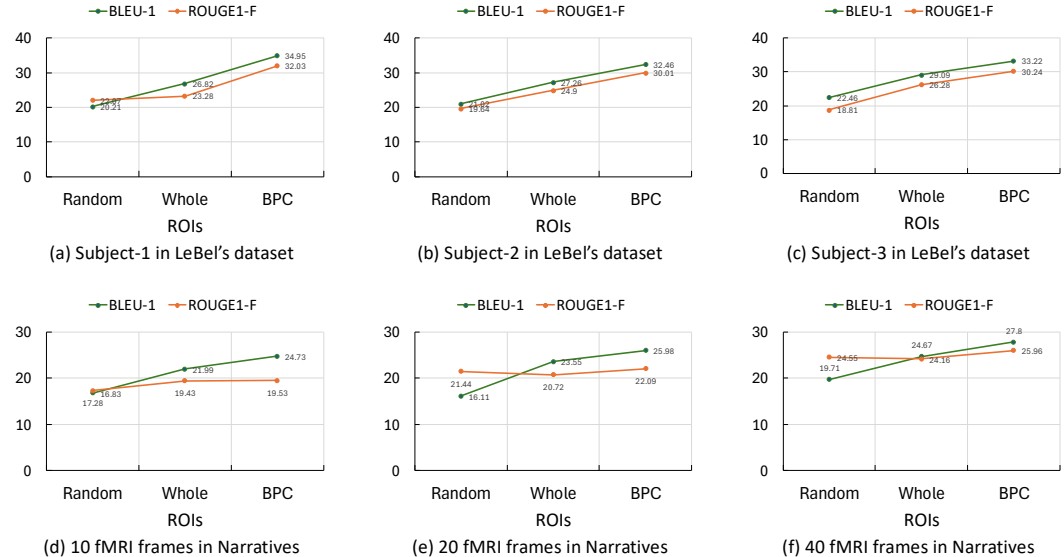

Figure 9: The performance of PREDFT with different ROIs selected in the side network. Complete results are shown in Table 10 and Table 11.

The results of cross-subject decoding in Narratives dataset are shown in Table 2. To further investigate the effect of fMRI sequence length on decoding model performance, experiments are separately conducted with fMRI sequence length of 10, 20, and 40, which equals to 15s, 30s, and 60s of fMRI with 1.5s TR in Narratives dataset. Despite UniCoRN's good performance in some evaluation metrics like ROUGE1-R, PREDFT achieves the best overall performance on all three experiments with different fMRI sequence lengths. Specifically, it achieves the highest BLEU-1 score of 27.8% on decoding 40 continuous fMRI frames. From the relatively low results of BLEU-2/3/4, we find all the models struggle to generate long accurate text. This indicates decoding continuous language accurately is still challenging. We don't observe significant differences in the impact of fMRI sequence length on the performance of the decoding models. Moreover, decoding on a within-subject basis generally yields better results than cross-subject decoding.

## 4.3 REGIONS OF INTERESTS SELECTION

To better understand whether PREDFT benefits from introducing brain predictive coding, we select different regions of interest (ROIs) for the side network to test their impacts on decoding performance. Aligned with previous analysis on predictive coding verification (see Section 2), three types of ROIs are selected: (a) "Random" means we randomly pick ROI area from the brain. (b) "Whole" means the whole human cerebral cortex is applied for the side network. (c) "BPC" denotes the ROIs related to brain predictive coding as verified in Section 2. It consists of Superior Temporal Sulcus (STS), Inferior Frontal Gyrus (IFG), Supramarginal Gyrus (SMG) and Angular Gyrus. BPC also covers most of the regions known for their significant role in language processing, like Auditory Cortex (AC), Prefrontal Cortex (PFC), and Broca area. The specific regions used in experiment are listed in Appendix A.4. Experiments are conducted in LeBel's dataset and Narratives dataset and results are illustrated in Figure 9. Figure 9 (a)-(c) display the decoding performance of three subjects from LeBel's dataset, and Figure 9 (d)-(f) show cross-subject decoding performance with different fMRI sequence lengths in the Narratives dataset. BLEU-1 and ROUGE1-F are selected to reflect the overall decoding performance under different settings.

Generally speaking, BPC area leads to the best performance on both datasets, while whole ROIs selection leads to sub-optimal decoding performance. However, random selection of ROIs results in poor decoding accuracy, both in the Narratives dataset with cross-subject decoding setting and LeBel's dataset with within-subject decoding setting. The experimental results are consistent with the findings in predictive coding verification. Two conclusions could be drawn from the above ROIs analysis. First, the predictive coding information can only be decoded from specific regions of the

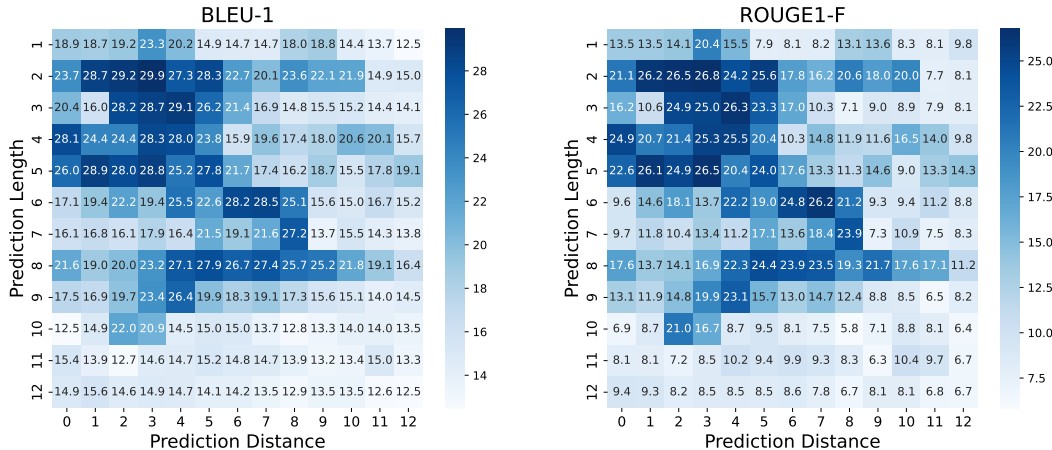

Figure 10: The impact of prediction length $l$ and distance $d$ on decoding performance. Results are averaged across three subjects in the LeBel's dataset. Per-subject results are shown in Figure 17, 18, and 19 respectively.

human brain. Second, brain predictive coding can be beneficial to fMRI-to-text models with proper network architecture design (e.g. our design of PREDFT).

### 4.4 PREDICTION LENGTH AND DISTANCE ANALYSIS

In this section, we investigate the impact of prediction length and prediction distance on the decoding performance of PREDFT. Same as the definition in Section 2, prediction length $l$ measures the length of continuous future predicted words. Prediction distance $d$ is the distance from the first word within each fMRI frame to the first future word. Experiments are conducted on LeBel's dataset and Narratives dataset. For the LeBel's dataset, decoding performance of all the three subjects with prediction length $l$ ranging from 1 to 12 and prediction distance $d$ ranging from 0 to 12 are tested. Figure 10 displays the average decoding performance of the three subjects. More results for each subject are presented in Appendix C.2. For the Narratives dataset, the prediction length is restricted as $l = 2$ and the prediction distance ranges from 0 to 10 due to the computational cost. We also try different fMRI sequence lengths under this setting and results are displayed in Appendix C.1. BLEU-1 and ROUGE1-F are chosen for evaluating fMRI-to-text decoding performance.

As shown in Figure 10, we observe a similar phenomenon as the predictive coding verification experiment in Figure 2. The information decoded from the process of brain predictive coding can improve the fMRI-to-text decoding performances, with a dependence on how far into the future humans are presumed to predict and how long the predicted content is. The decoding performance of PREDFT first rises then falls as prediction distance $d$ increases for most prediction lengths. For a short (e.g. $l = 2, 3$) or long (e.g. $l = 9, 10$) prediction length, the rising point of decoding performance comes earlier compared to a medium prediction length (e.g. $l = 6, 7, 8$). An inappropriate prediction length, whether excessively short (e.g. $l = 1$) or long (e.g. $l = 11, 12$), will result in poor performance. This is somewhat different from the predictive coding verification where short or long prediction length will still contribute to the increment of prediction score marginally.

## 5 RELATED WORK

**fMRI-to-text Decoding.** Most existing studies focused on aligning fMRI signal to a limited vocabulary of items and performing word-level decoding (Bhattasali et al., 2019; Wang et al., 2020; Affolter et al., 2020; Zou et al., 2021), or sentence-level classification (Pereira et al., 2018; Sun et al., 2019). Recently, researchers turned to powerful pre-trained language models for open-vocabulary fMRI-to-text decoding. For example, Tang et al. (2023) designed a pipeline model where the encoder is responsible for identifying the most possible word sequence among candidates generated by the GPT model with beam search. Zhao et al. (2024) further improved Tang's method by applying contrastive learning to pre-train an fMRI-text mapper. Xi et al. (2023) proposed a three-phase training

framework UniCoRN which applies BART (Lewis et al., 2020) model for generation. Ye et al. (2023) proposed BrainLLM by concatenating fMRI embedding with word embedding as input prompt to fine-tune a Llama2 (Touvron et al., 2023) model.

**Brain Predictive Coding.** Predictive coding theory (McClelland & Rumelhart, 1981; Rao & Ballard, 1999; Friston & Kiebel, 2009) aims to propose a potential unifying theory for computational and cognitive neuroscience (Millidge et al., 2021). It was initially proposed as a neuroscientific theory (Mumford, 1991) and subsequently developed into its mathematical form of cortical responses (Friston, 2008). Although originally formulated to investigate brain visual processing, it was also extended to language processing in the human brain in previous work (Garrido et al., 2009; Wacongne et al., 2011). Predictive coding suggests that human brain naturally makes predictions about future words and sentences when it perceives natural language stimuli. Such hypothesis has already been evidenced by correlating word or phonetic surprisal with fMRI or EEG (Willems et al., 2016; Okada et al., 2018; Donhauser & Baillet, 2020; Heilbron et al., 2022). Caucheteux et al. (2023) further verified a predictive coding hierarchy in the human brain listening to speech by investigating the linear mapping between modern language models and brain responses.

## 6 DISCUSSION

This paper explores the integration of predictive coding theory into model design for decoding fMRI signals into natural language. First, we analyze the effect of brain predictive coding by linearly mapping the activations of the computational auto-regressive language model to brain responses. Then we verify those effects across different temporal and spatial scales. Motivated by the observations, an fMRI-to-text decoding model PREDFT is proposed, which utilizes a side network to capture and fuses brain prediction into the language reconstruction process. Comprehensive experiments demonstrate the superior decoding performance of PREDFT benefits from integrating brain predictive coding.

While existing studies have successfully mapped the representations drawn from auto-regressive language models with brain responses to auditory language stimuli (Tang et al., 2023), the reasons behind this success are still controversial. A possible explanation is that both the language models and humans follow a next-word prediction pattern while learning language-related knowledge. However, Antonello & Huth (2024) questioned this hypothesis and claimed that language models can be used for predicting brain responses because they generally capture a wide variety of linguistic phenomena. Based on the existing analysis of the predictive coding theory, we further explored the potential of applying predictive coding heuristics into fMRI-to-text decoding. We show the effect of predictive coding on language decoding in different ROIs, prediction lengths, and prediction distance. This finding provides a novel view of the temporal and spatial scales in predictive coding.

Non-invasive neural decoding is an emerging research topic. PREDFT fuses brain predictive coding in language reconstruction. One disturbing fact is that human not always predict the right future words, which might become distraction in the decoding process. Despite the improvement in decoding performance in PREDFT, we find it's still challenging to reconstruct natural language from fMRI signals. The challenges can be summarized as follows: First, the noise inherited from collecting fMRI data is a natural barrier to decoding. Second, different from fMRI-to-image decoding (Wang et al., 2024; Scotti et al., 2024) whose experimental setting is requiring subjects look at pictures one by one with certain intervals, the fast spoken word rate isn't compatible with the low temporal resolution of fMRI data in fMRI-to-text decoding. So part of the brain responses are not recorded in fMRI. Such a hypothesis has been evidenced through experiments in Appendix D. Building a dataset with high temporal resolution devices may alleviate this problem. Considering the quality of current naturalistic language comprehension fMRI dataset (mostly building upon 3T scanner), we think it's better to change the evaluation standard from word-level to semantic-level, as reflected in case study.

The limitations of this work include: (i) Experiments are only conducted on fMRI datasets, i.e., LeBel's dataset, Narratives. Exploration of other experimental setups (e.g. visual stimuli Pereira et al. (2018)) and different modalities of signals (e.g. magnetoencephalogram (MEG)) is an emerging direction. (ii) Contents that are not expected by the subjects might make it difficult for the brain predictive coding function to decode. We leave it as future work to analyze this effect.

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

# A EXPERIMENTAL SETTINGS

## A.1 BASELINE METHODS

In this part, we briefly introduce the compared baseline methods in experiments.

- Tang's method (Tang et al., 2023): The model first applies an encoding model to predict how subject's brain responds to natural language. The linear encoding model capture the simulate the brain responses to stimulus. A GPT model with beam search decoding algorithm is applied in decoding fMRI signal to text. The beam contains $k$ most likely candidate word sequences, and the sequence with most similarity to the encoded brain signal is chosen as final generated content.

- MapGuide (Zhao et al., 2024): This model improves Tang's method on the fMRI encoding process by applying contrastive learning. Specifically, the MapGuide framework follows a two-stage fMRI-to-text decoding manner. The stage-A pre-trains a robust fMRI-to-text mapper. The mapper first optimize Mean Squared Error loss between predicted text and ground truth, and then optimize the infoNCE loss between original fMRI representation and masked fMRI representation. The stage-B is similar to Tang's model. A GPT model (Radford et al., 2018) generates candidate word through beam search and the highest possible word is chosen based on the similarity between projected text embedding and GPT output.

- BrainLLM (Ye et al., 2023): This approach tackles the task of decoding text from functional magnetic resonance imaging (fMRI) data by employing an auto-regressive generation method. It involves using the decoded representations from fMRI directly as inputs for a large language model (LLM). This method eliminates the dependency on the accuracy of pre-constructed candidate phrases, thereby enhancing the fidelity and robustness of text generation from brain activity data. In practice, we try Llama-2 (Touvron et al., 2023) for the large language model in generation.

- UniCoRN (Xi et al., 2023): UniCoRN provides a unified encoder-decoder framework for EEG and fMRI to text decoding. The training of UniCoRN follows a three-stage manner. The fMRI encoder is first pre-trained with a cognitive signal reconstruction task to capture spatial feature. Then a Transformer encoder is stacked into the fMRI encoder to capture temporal connections. Finally BART (Lewis et al., 2020) is fine-tuned to translate fMRI representation into natural language in the generation stage.

### A.2 DATASETS AND SPLITTING METHODS

Experiments are conducted on two popular naturalistic language comprehension fMRI datasets. The Narratives (Nastase et al., 2021) is currently the largest naturalistic language comprehension fMRI dataset, containing recordings from 345 subjects listening to 27 diverse stories. Since the data collection process involves different machines, only fMRI data with $64 \times 64 \times 27$ voxels is considered, which leads to 230 subjects. The predictive coding verification applies the AFNI-nonsmooth pre-processing method (2D brain surface data). For the fMRI-to-text decoding experiment, fMRIPrep version (4D whole brain data) is selected to follow the settings in Xi et al. (2023). The LeBel's dataset (LeBel et al., 2023) contains eight subjects participating a passive natural language listening task. Following Tang's setting (Tang et al., 2023), only subject-1, subject-2, and subject-3 are applied in both predictive coding verification and fMRI-to-text decoding experiment (2D brain surface data).

How to split datasets for training and evaluation is a matter of debate in fMRI-to-text decoding (Xi et al., 2023). Generally speaking, dataset splitting can be categorized into two main approaches: **within-subject** splitting and **cross-subject** splitting. Under the within-subject splitting setting, fMRI signal and text pairs $\langle F_{i,j}, U_j \rangle$ of training, validation, and test set all comes from one subject, namely $i$ is fixed. While in cross-subject data splitting, fMRI signal comes from different test subjects, i.e., $i$ is not fixed for training, validation, and test set. Ye et al. (2023); Tang et al. (2023); Zhao et al. (2024) trained and evaluated models within subject in LeBel's dataset. Xi et al. (2023) applied cross-subject splitting in Narratives dataset, but has been identified to have data leakage issue (Yin et al., 2023). To avoid data leakage and test model's cross-subject generalization ability, we apply the splitting method proposed in Yin et al. (2023), which follows two rules in the dataset splitting process: (i) fMRI signals collected from specific subject in validation set and test set will not appear in training set, which means the trained encoder cannot get access to any brain information belonging to subjects in test or validation set. (ii) Text stimuli in validation set and test set will not appear in training set.

### A.3 EVALUATION METRICS

- BLEU (Papineni et al., 2002): BLEU (Bilingual Evaluation Understudy) is an algorithm for evaluating the quality of text which has been machine-translated from one natural language to another. Quality is considered to be the correspondence between a machine's output and ground truth label. Neither intelligibility nor grammatical correctness are not taken into account. BLEU is calculated in the following way. The geometric average of the modified n-gram precisions $p_n$ are first computed, with $n$-gram up to length $N$ and positive weight $w_n$ summing to one. The brevity penalty BP is computed through

$$\text{BP} = \begin{cases} 1 & \text{if } c > r \\ e^{(1-r/c)} & \text{if } c \leq r \end{cases} \tag{7}$$

where c is the candidate translation lenght and r is the effective reference corpus length. Then the BLEU score is calculated.

$$\text{BLEU} = \text{BP} \cdot \exp\left(\sum_{n=1}^{N} w_n \log p_n\right) \tag{8}$$

The maximum $N$ is set as 4 with $w_n = 1/4$, corresponding to the BLEU-4 score.

- ROUGE (Lin, 2004): ROUGE (Recall-Oriented Understudy for Gisting Evaluation) is a suite of metrics often employed to evaluate the quality of automatic text summarization and machine translation in natural language processing (NLP). It assesses similarity by comparing machine-generated content against one or more reference texts. ROUGE scores range from 0 to 1, with 1 indicating the highest level of similarity. Specifically, ROUGE-Precision measures the accuracy of the machine-generated content by assessing how closely it matches the reference content in terms of

Table 3: Notation Table of different symbols in Methodology.

| Symbol | Definition |
| --- | --- |
| $\mathcal{D}$ | Naturalistic language comprehension fMRI dataset |
| $U_j$ | The $j$-th input text stimuli |
| $F_{i,j}$ | Input fMRI image of $i$-th subject hearing text stimuli $U_j$ |
| $V_j$ | The $j$-th input predicted word sequence extracted from $U_j$ |
| $R_{i,j}$ | Predictive coding related ROIs extracted from $F_{i,j}$ |
| $\mathcal{M}_{\theta,\phi}$ | The PREDFT model |
| $\mathcal{M}_{\theta}$ | The main network of PREDFT |
| $\mathcal{M}_{\phi}$ | The side network of PREDFT |
| $\mathcal{M}_{\theta_{\mathrm{Enc}}}$ | The encoder in the main network |
| $\mathcal{M}_{\theta_{\mathrm{Dec}}}$ | The decoder in the main network |
| $\mathcal{M}_{\phi_{\mathrm{Enc}}}$ | The encoder in the side network |
| $\mathcal{M}_{\phi_{\mathrm{Dec}}}$ | The decoder in the side network |
| $H^P_{\theta_{\mathrm{Enc}}}$ | The output of $\mathcal{M}_{\theta_{\mathrm{Enc}}}$ |
| $H^Q_{\theta_{\mathrm{Dec}}}$ | The output of $\mathcal{M}_{\theta_{\mathrm{Dec}}}$ |
| $H^M_{\phi_{\mathrm{Enc}}}$ | The output of $\mathcal{M}_{\phi_{\mathrm{Enc}}}$, also the input of $\mathcal{M}_{\theta_{\mathrm{Dec}}}$ |

content. A higher precision score indicates that the machine-generated content includes a significant portion of relevant information from the reference content, while minimizing the inclusion of extraneous or irrelevant details. ROUGE-Recall measures the extent to which a machine-generated content captures the information contained in a reference content. It is particularly useful for assessing how much of the key content from the reference is retained by the machine-generated output. A higher recall score suggests that the model has effectively captured a significant portion of the reference information. However, it is important to note that a high recall value may sometimes indicate the inclusion of redundant information, which could potentially lead to a decrease in precision. ROUGE-F1 helps in maintaining this balance by combining both precision and recall into a single value.

### A.4 IMPLEMENTATION DETAILS

**Cortical Parcellation.** For the LeBel's dataset, we apply the cortical parcellation provided by Tang et al. (2023). "Auditory" region is applied for the BPC region in PREDFT. For the random ROIs selection, we randomly choose 1000 voxels from brain surface data. For the Narratives dataset, the latest version of Destrieux atlas (Destrieux et al., 2010) is applied for cortical parcellation, which leads to 74 regions per hemisphere. We use six regions of interests that have been proven in (Caucheteux et al., 2023) to contribute to brain prediction, including superior temporal sulcus, angular gyrus, supramarginal gyrus, and opercular, triangular, orbital part of the inferior frontal gyrus. In the ROIs selection experiment, G_and_S_cingul-Ant, G_and_S_subcentral, G_and_S_transv_frontopol, G_orbital, S_front_middle, S_subparietal are selected in random ROIs experiment.

**Hyper-parameters.** For the predictive coding verification experiment, we use 'RidgeClassifierCV' regressor from scikit-learn (Pedregosa et al., 2011) to predict the continuous features and align language models to brain, with 10 possible penalization values log-spaced between $10^{-1}$ and $10^8$. The linear model is evaluated on held out data, using 10 cross-validation for brain score of each subject. In practice, Principal Component Analysis (Abdi & Williams, 2010) is applied to reduce the dimension of GPT-2 output (768) to 20. The output of eighth layer in GPT-2 is applied as activation.

4D volumetric whole brain data in Narratives dataset and 2D brain surface data is applied for the fMRI-to-text decoding experiment. For 4D brain data, voxel-level normalization is first performed to raw 4D fMRI data, which separately normalizes the values of each voxel over the time domain. This normalization highlights the relative activation of a specific voxel within given intervals. In the main decoding network, the 3D-CNN module contains $L = 18$ layers. The numbers of Transformer encoders and decoders are set to $P = 4$ and $Q = 12$ respectively. For the side network, both are set as $M = N = 6$. We apply the BART(Lewis et al., 2020) tokenizer for the Narratives dataset, and the tokenizer provided in Tang et al. (2023) for the LeBel's dataset. PREDFT is trained from

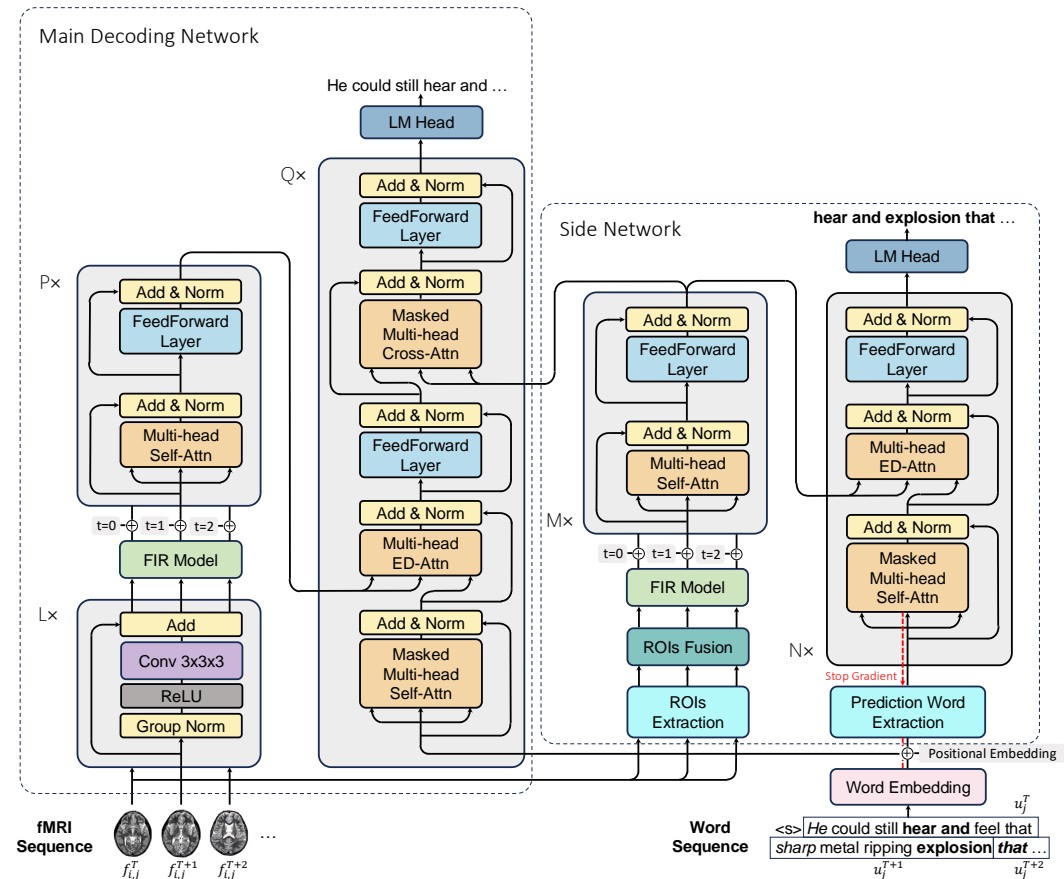

Figure 11: The general framework of PREDFT. The *italic* words in the input word sequence stand for the first heard word of each fMRI image while the **bold** words stand for the prediction words.

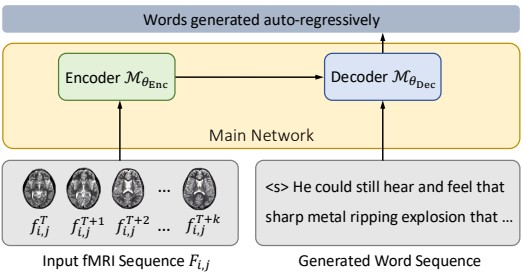

Figure 12: The illustration of PREDFT without SideNet

scratch with 40 epochs and the initial learning rate is set as 5e-4 which eventually decays to 1e-5. The hyper-parameter $\lambda$ for jointly training the main and side networks is set to 1 for fMRI sequence of length 10, and 0.5 for sequence of length 20 and 40 in Narratives dataset. For baseline methods we strictly follow the settings in the proposed paper. All experiments are conducted on NVIDIA A100-80G GPUs. The total parameters for PREDFT is around 200 million. The time complexity of PREDFT is the same as vanilla Transformer, which is $\mathcal{O}(lhn^2)$, where $n$ is the input length of word sequence, $l$ is the batch size, $h$ is the number of attention heads.

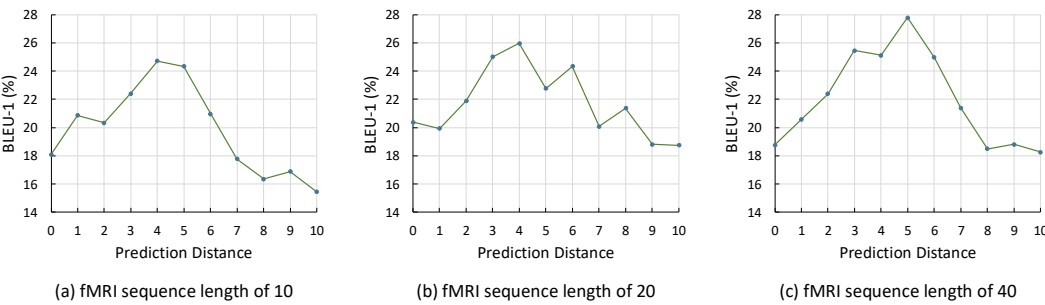

(a) fMRI sequence length of 10

(b) fMRI sequence length of 20

(c) fMRI sequence length of 40

Figure 14: The impact of prediction distance on decoding performance in Narratives dataset.

## B    DETAILED ILLUSTRATION OF PREDFT

In this section, we present the detailed framework of PREDFT in Figure 11 and illustration of PREDFT without SideNet which is used as compared baseline in experiments in Figure 12. We also make a notation table 3 for the symbols mentioned in the Methodology part.

## C    SUPPLEMENTARY EXPERIMENTS

### C.1    EXPERIMENTS ON THE NARRATIVES DATASET

Two kinds of experiments on the Narratives dataset are presented in this section. For the predictive coding verification, 230 subjects in the Narratives dataset are selected for the experiment. The brain score is averaged across subjects and computed within one fMRI frame, namely $N$ is set as 1. The output of eighth layer in GPT-2 is applied as activation and we always choose the activation of the first word within each fMRI frame as $X$. Due to the high computational cost of this experiment, we don't test changing prediction length and prediction distance like we try in the LeBel's dataset. Instead, the prediction length $l$ is set as 2 and the prediction distance $d$ ranges from 0 to 11.

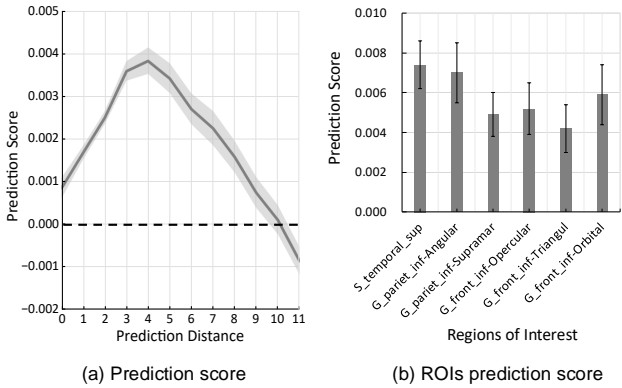

(a) Prediction score

(b) ROIs prediction score

Figure 13: Predictive coding verification on the Narratives dataset with prediction length $l = 2$.

Figure 13(a) reports the prediction score across individuals with $95\%$ confidence intervals. Results show prediction score $P_{(d,l)}(X)$ first increases and peaks at $d = 4$, then decreases as the prediction distance $d$ increases, and finally goes down below zero when the prediction distance comes to $d = 11$. We also conduct regions of interest (ROIs) analysis. Six regions related to brain predictive coding, including superior temporal sulcus, angular gyrus, supramarginal gyrus, and opercular, triangular, and orbital part of the inferior frontal gyrus in the left hemisphere, are selected for experiments. Sub-figure (b) shows the prediction score of six ROIs across individuals with $95\%$ confidence intervals. The prediction distance is set at $d = 4$ for the best predictive performance, as reflected in sub-figure (a). All the selected ROIs show positive responses to prediction words.

For the fMRI-to-text decoding experiment, similar to the prediction length and distance experiment on LeBel's dataset, we analyze the changing of prediction length and distance to the decoding performance on the Narratives dataset. Experiments are conducted with a fixed window length $l = 2$ and the influence of prediction distance to decoding performance is reflected through BLEU-1 score. Figure 14 shows the results with different fMRI sequence lengths. We notice a similar phenomenon

as the predictive coding verification experiment. The trend of BLEU-1 score first rises then falls as prediction distance $d$ increases. PREDFT achieves the best performance with $d$ around 4.

## C.2 Experiments on the LeBel's Dataset

We analyze the influence of prediction length and distance per subject. Figure 17, 18 and 19 show results on subject-1, subject-2, subject-3 respectively. All the three subjects show very similar trends in the changing of decoding performance. Align with the conclusions in Section 4.4, despite occasional fluctuations, the decoding performance first rises and then falls when prediction distance extends. Moreover, the best performance comes with a medium prediction length and distance.

## D Decoding Error Analysis

In this section, we design an experiment that analyzes the positional distribution of incorrectly decoded words. Figure 15 is an example of two successive fMRI frames containing seven and four spoken words respectively. Three kinds of errors during decoding are defined: (i) The decoding model $\mathcal{M}$ fails to generate the correct word (e.g. "could" is incorrectly decoded as "should"). (ii) The decoding model $\mathcal{M}$ generates redundant words (e.g. repetitive "and"). (iii) Corresponding words are missing during decoding (e.g. "sharp metal ripping" is missing

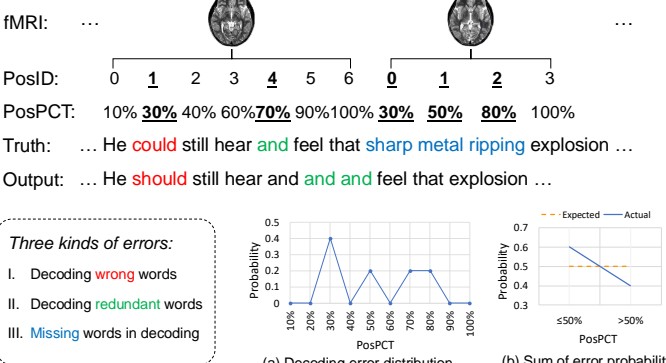

Figure 15: An example of the experiment for decoding error analysis. PosID and PosPCT stand for the word position index of truth and the percentage of index respectively.

in output). Although some of the generated words are semantically consistent with the ground truth, we apply the strict exact match to facilitate automatic evaluation. Three position counting methods for corresponding errors are proposed: (i) If decoded word is wrong, position index of the corresponding truth word is marked as wrong. (ii) If decoded words are redundant, position index of the last matched truth word is marked as wrong. (iii) If decoded words are missing, position indices of all the missing truth words are marked as wrong. Since different fMRI frames contain different numbers of spoken words, the relative positions of incorrect words within each frame, namely the percentage of index (PosPCT), are considered. The error probability of one specific position is the proportion of errors at this position to the total number of errors. Table (a) in Figure 15 illustrates the positional distribution of incorrectly decoded words in the example. The distribution is calculated at ten percentiles from $10\%$ to $100\%$. As some positions like $10\%$ or $90\%$ are minority in all statistical positions, we also add the error probabilities of the first and last $50\%$ respectively, as shown in table (b).

Such experiment is conducted on UniCoRN and PREDFT. Positional decoding error distribution and sum of error probability are analyzed. Results are shown in Figure 16. We find the error probability of last heard words in TR is significantly higher than words heard at the beginning. However, the error probabilities of decoding the first and last half of text are supposed to be the same in normal cases. This phenomenon leads to the hypothesis that the information of some heard words, especially the last few words in each TR, is lost in fMRI data. It's caused by the discrete sampling feature of fMRI: Due to the constraints of MRI scanner in strength and speed of switching the magnetic gradients, the fMRI signal is sampled discretely with a fixed time interval called repetition time (TR) in order to achieve the balance between spacial and temporal resolution. The repetition time in fMRI-to-text decoding task is usually around two seconds. However, the average speaking rate of human is about three words per second. If pauses between sentences are excluded, the word rate within one sentence will increase to five per second. This feature of fMRI leads to the information loss problem in fMRI-to-text decoding: While brain responses of the first few heard words are recorded in one fMRI frame, information of the last heard words is lost due to the low temporal resolution,

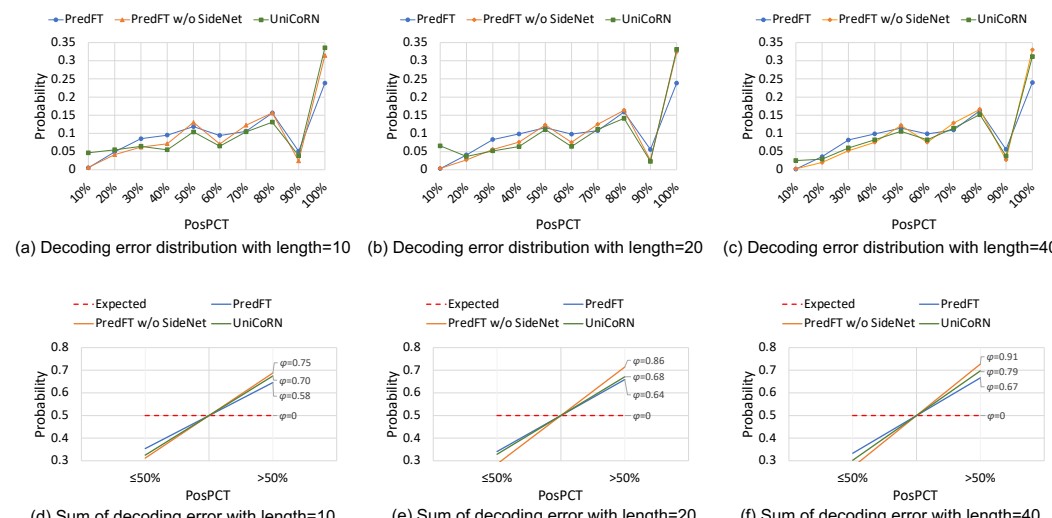

Figure 16: Information loss of different models under different fMRI sequence lengths in Narratives.

Table 4: Decoding performance of PREDFT under different hyper-parameter $\lambda$ in Narratives.

| Length | $\lambda$ | BLEU-1 | BLEU-2 | BLEU-3 | BLEU-4 | ROUGE1-R | ROUGE1-P | ROUGE1-F |
|---|---|---|---|---|---|---|---|---|
| 10 | 1 | **24.73** | **8.39** | **3.92** | **1.86** | 14.07 | **35.28** | **19.53** |
| | 0.75 | 22.32 | 4.44 | 0.87 | 0.12 | **16.57** | 19.77 | 17.96 |
| | 0.5 | 22.56 | 4.59 | 1.26 | 0.41 | 15.84 | 19.54 | 17.44 |
| | 0.25 | 21.13 | 5.21 | 1.26 | 0.35 | 14.00 | 26.67 | 18.25 |
| 20 | 1 | 18.33 | 5.00 | **1.37** | **0.48** | 15.60 | **31.97** | 20.90 |
| | 0.75 | 21.15 | 4.71 | 1.22 | 0.44 | **20.58** | 27.13 | **23.35** |
| | 0.5 | **25.98** | **5.61** | 1.36 | 0.21 | 19.61 | 25.43 | 22.09 |
| | 0.25 | 25.21 | 5.59 | 1.35 | 0.24 | 20.46 | 26.24 | 22.95 |
| 40 | 1 | 20.56 | 5.20 | 1.24 | 0.26 | 21.92 | 28.74 | 24.82 |
| | 0.75 | 26.73 | 7.13 | 1.55 | 0.49 | 19.21 | 31.17 | 23.72 |
| | 0.5 | **27.80** | **8.29** | **2.00** | **0.54** | 19.53 | **38.95** | **25.96** |
| | 0.25 | 20.28 | 4.73 | 0.84 | 0.21 | **22.12** | 28.40 | 24.82 |

making decoding these words difficult. The latency of BOLD signal complicates the theoretical explanation of this phenomenon. But based on experimental results and previous study (Liao et al., 2002) which indicates the latency of fMRI response is about six seconds, exactly an integer multiple of repetition time (1.5s or 2s), the hypothesis of information loss is reasonable.

From sub-figure (a), (b), (c) in Figure 16, we surprisingly find PREDFT successfully reduces the error probability of the last few decoded words compared to UniCoRN. This implies the predictive coding information in brain could be utilized to alleviate the information loss, and such alleviation of information loss is closely related to the decoding accuracy. To better illustrate the degree of information loss, we propose a novel index *information loss slope* $\varphi$ measuring the growth rate of error probability from the first half of decoded content to the last half,

$$\varphi = \frac{\sum_{i=6}^{10} p_i - \sum_{i=1}^{5} p_i}{0.5}, \tag{9}$$

where $p_i$ stands for the error probability of $i\%$ position. $\varphi$ is expected to be around zero, as the error probabilities of different positions are supposed to be the same without information loss. However, the $\varphi$ values of all compared models are high, indicating that all models suffer from information loss. PREDFT successfully mitigates information loss to some extent. As shown in sub-figure (d), (e), (f) of Figure 16, the $\varphi$ score of PREDFT is lower than compared models on all the three experiments with different fMRI sequence lengths.

Table 5: Decoding performance of PREDFT under different hyper-parameter $\lambda$ in LeBel's dataset.

| | $\lambda$ | BLEU-1 | BLEU-2 | BLEU-3 | BLEU-4 | ROUGE1-R | ROUGE1-P | ROUGE1-F |
|---|---|---|---|---|---|---|---|---|
| **Sub-1** | 1 | 34.95 | **14.53** | **5.62** | **1.78** | 23.79 | 49.95 | **32.03** |
| | 0.75 | 27.26 | 10.07 | 3.47 | 1.47 | 16.10 | **54.56** | 24.75 |
| | 0.5 | **34.97** | 13.43 | 4.68 | 1.42 | **24.42** | 43.72 | 31.21 |
| | 0.25 | 27.32 | 9.61 | 3.13 | 0.00 | 16.22 | 53.38 | 24.64 |
| **Sub-2** | 1 | **32.46** | **11.77** | 3.95 | 0.84 | **24.90** | 38.43 | **30.01** |
| | 0.75 | 20.21 | 7.25 | 2.64 | 0.62 | 13.89 | 55.16 | 22.07 |
| | 0.5 | 19.23 | 6.96 | 2.28 | 0.55 | 12.80 | **62.37** | 21.09 |
| | 0.25 | 30.33 | 11.28 | **4.02** | **1.55** | 20.31 | 40.82 | 26.93 |
| **Sub-3** | 1 | **33.22** | **12.91** | **4.29** | **1.76** | 23.22 | **44.31** | **30.24** |
| | 0.75 | 33.17 | 11.06 | 2.97 | 0.00 | **25.84** | 34.75 | 29.49 |
| | 0.5 | 31.89 | 11.08 | 3.54 | 1.15 | 24.22 | 35.59 | 28.63 |
| | 0.25 | 29.62 | 10.18 | 3.09 | 0.70 | 20.05 | 43.23 | 27.15 |

# E ABLATION STUDY

Four aspects of ablation experiments are conducted to analyze PREDFT. First we test whether the side network for brain prediction really improves decoding accuracy. The model PREDFT without side network (PREDFT w/o SideNet) is built with the same settings as PREDFT during training except for only keeping the main decoding network (the cross-attention layers in main decoding network are removed). The results of this model's decoding performance are listed in Table 1 and Table 2, same as not using ROIs in side network ("None" in the table). The performance of PREDFT w/o SideNet is significantly worse than PREDFT in all the three experiments with different fMRI sequence length. It also performs worse than UniCoRN under most cases, which might be attributed to the pretrained language model used in UniCoRN. For the three test subjects in LeBel's dataset, the decoding accuracy without side network also gets severe decrement.

Besides, the decoding error distribution of PREDFT w/o SideNet is counted to verify whether the side network helps alleviate information loss. As shown in Figure 16, the error distribution of PREDFT w/o SideNet across different positions is similar to that of UniCoRN. The probability of decoding error increases as the word position moves backward within one fMRI frame, peaking at the position of the last word. PREDFT w/o SideNet severely suffers from information loss as shown in sub-figure (d), (e), (f) of Figure 16, with the highest information loss slope. The ablation experiments provide solid evidence on the effectiveness of the side network in PREDFT.

We also test the influence of hyper-parameter $\lambda$ to decoding performance of PREDFT. As shown in Table 4 and Table 5, four different $\lambda$ values ranging from 0.25 to 1 are tested in experiments with different fMRI sequence lengths (the Narratives dataset) and different subjects (the LeBel's dataset). Empirically, PREDFT achieves relatively good decoding accuracy with $\lambda = 0.5$ and $\lambda = 1$ in all experiments.

Finally, we conduct a chance-level experiment to verify that our model learns language reconstruction from fMRI response instead of random signals. Specifically, we randomly shuffle the order of the input fMRI images, and maintain all other hyper-parameters. The results are shown in Table 6 and Table 7. We notice that chance-level PREDFT performs extremely poor.

Table 6: The performance of different models in within-subject fMRI-to-text decoding in LeBel's dataset. 10 continuous fMRI images (equals to 20 seconds) are sampled for decoding.

| | Models | BLEU-1 | BLEU-2 | BLEU-3 | BLEU-4 | ROUGE1-R | ROUGE1-P | ROUGE1-F | BERTScore |
|---|---|---|---|---|---|---|---|---|---|
| **S1** | Chance-level PREDFT | 20.34 | 3.75 | 0.20 | 0 | 15.48 | 20.41 | 17.45 | 77.7 |
| | PREDFT | **34.95** | **14.53** | **5.62** | **1.78** | **23.79** | **49.95** | **32.03** | **82.92** |
| **S2** | Chance-level PREDFT | 18.96 | 2.96 | 0 | 0 | 15.04 | 20.37 | 17.18 | 78.02 |
| | PREDFT | **32.46** | **11.77** | **3.95** | **0.84** | **24.90** | **38.43** | **30.01** | **82.52** |
| **S3** | Chance-level PREDFT | 19.48 | 3.58 | 0.28 | 0 | 15.17 | 19.35 | 16.96 | 78.24 |
| | PREDFT | **33.22** | **12.91** | **4.29** | **1.76** | **23.22** | **44.31** | **30.24** | **82.11** |

Table 7: The performance of different models in cross-subject fMRI-to-text decoding in Narratives dataset. Length denotes the length of time windows for continuous fMRI frames.

| Length | Models | BLEU-1 | BLEU-2 | BLEU-3 | BLEU-4 | ROUGE1-R | ROUGE1-P | ROUGE1-F | BERTScore |
|---|---|---|---|---|---|---|---|---|---|
| 10 | Chance-level PREDFT | 19.92 | 2.60 | 0 | 0 | 13.28 | 22.80 | 16.72 | 76.23 |
| | PREDFT | 24.73 | 8.39 | 3.92 | 1.86 | 14.07 | 35.28 | 19.53 | 78.52 |
| 20 | Chance-level PREDFT | 19.53 | 2.45 | 0 | 0 | 14.94 | 21.55 | 17.58 | 75.39 |
| | PREDFT | 25.98 | 5.61 | 1.36 | 0.21 | 19.61 | 25.43 | 22.09 | 78.20 |
| 40 | Chance-level PREDFT | 20.31 | 2.88 | 0.41 | 0 | 15.39 | 24.76 | 18.80 | 75.58 |
| | PREDFT | 27.80 | 8.29 | 2.00 | 0.54 | 19.53 | 38.95 | 25.96 | 78.63 |

# F CASE STUDY

Some fMRI-to-text decoding cases are analyzed in this section. We show cases from Narratives dataset and LeBel's dataset with fMRI sequence length 10. Some of the selected samples are displayed in Table 8 and Table 9 respectively.

Despite the relatively good automatic evaluation performance, all the models struggle to decode generally accurate content, especially in (i) generating fluent and coherent sentences. During the experiment we observed that two types of fMRI-to-text decoding methods encounter different problems. The Bayesian decoding method (e.g. Tang's model, MapGuide) is able to generate grammatically fluent sentences. However, it's hard to decode correct semantic information. While fine-tuning method (e.g. UniCoRN, PREDFT) sometimes struggle to decode fluent sentences, it can also decode some key concepts from fMRI signals. (ii) capturing fine-grained semantic meanings (e.g. "jealous of her", "don't have my driver's license") (iii) decoding specific terminology (e.g. name "Mary", location "florida") or complicated phrases (e.g. "sharp metal ripping explosion"). We find PREDFT successfully decodes some high-level semantic concepts and key words. For example, as shown in the bold words in Table 8, PREDFT conveys the meaning of "the best dream" while UniCoRN fails to in case1. In case2, PREDFT decodes the meaning of "he and Mary prepare to sleep". Generally speaking, PREDFT performs better than UniCoRN. More cases are shown in Table 9. PREDFT decodes the semantic of some key phrases in case 1-4. Case 5 is a bad case, where the model generates too much repetitve and wrong words.

Table 8: Cases of decoded content in Narratives dataset. **Bold** words indicate key phrases.

| | |
|---|---|
| Case1 | Truth: It was **more real than any dream** he had ever had in his life. He could still **hear and feel** that sharp metal ripping **explosion** that searing wave of heat. He **sat** |
| | UniCoRN: It's than I just a said of a a the hand. You have I the the the you I I I in to at the to to **sit**. |
| | PREDFT: It's a **more than normal** just good **dream** about to said. He open the eyes I have be her I like she and and and he and of under the next and I look the platform |
| Case2 | Truth: He couldn't shake the **thought** out of his mind. It persisted all through the day **until dinner**. He was still brooding as **he and Mary** got **ready for bed**. **Guy dear**. Hm oh no. **Anything wrong** |
| | UniCoRN: And I know a Dean and a eyes it. And of the first and the and a to **guy** him. I said I to to him out and And to in that in the end. |
| | PREDFT: He don't know my girl you of the eyes but **his girl sleep he** and he said and he said and the to the and and which I **not wrong**. But the **Guy** |

Table 9: Cases of decoded content in LeBel's dataset. **Bold** words indicate key phrases.

| | |
|---|---|
| Case1 | Truth: stories about our lives we're **both from up north** we're both kind of to the neighborhood this is in **florida** we both **went to college** not great colleges but man we graduated and i'm actually finding myself a little **jealous of her** because she has this really **cool job washing dogs** |
| | PREDFT: and i was well no she were close to we and **our family** gonna things that she gonna that of **we're the neighborhood** i **spend the time in the college** and were to the and gonna to i were not in our and **a little bit her and it was a amazing** and she |
| Case2 | Truth: it was **silent and lovely** and there was **no sound** except for ch ch ch ch ch ch ch ch and i was **enjoying myself** and enjoying the absence of anger and enjoying these few hours i knew i'd have of |
| | PREDFT: it was **in not sound** and it was a way like it the and that that that it and it that that that i was able to **enjoy i** to months and was **happy** that never back to the |
| Case3 | Truth: and we **start walking** and uh we get to this um lots of uh **lights and uh the roads** are getting wider and wider and there's **more cars** and i see um lots of **stores** you know and **dollar stores** and and then we cross over us |
| | PREDFT: and we were to **walk around** and we were to spend the time and we were ready what i what i **see a store** and i was come to the end of the store i know and i and you were the and |
| Case4 | Truth: and um i **don't have a baby** you know so i can leave whenever i want **i smoked all seven cigarettes on the way home** and people who have never smoked cigarettes just think disgusting and but unless you've had them and held them dear |
| | PREDFT: and um i and i know **a lot girl** know what i was do and **i have no children** to the time and i have been a lot time to think and **i smoked cigarettes** and i have to ever to me to life |
| Case5 | Truth: i **get home** and how sweet that'll be we are chain smoking off each other oh that's almost out come on and we we go through this entire pack until it's gone and then i say you know what uh this is a little funny but you're gonna have to show me **the way to get home** because although **i'm twenty three years old** i **don't have my driver's license** yet and i just jumped out |
| | PREDFT: i was **to home** and were able to each other and my what like of to the i were were to the time life and you like to i i am to know i i i time that lot bit girl i not be to do of to way and do a for **i gonna five hundred old** was know a to that i was a to of |

Table 10: The performance of PREDFT when different ROIs are selected for the side network under within-subject decoding setting in LeBel's dataset.

| | ROIs | BLEU-1 | BLEU-2 | BLEU-3 | BLEU-4 | ROUGE1-R | ROUGE1-P | ROUGE1-F |
|---|---|---|---|---|---|---|---|---|
| **Sub-1** | None | 27.91 | 10.26 | 3.50 | 1.29 | 18.59 | 49.00 | 26.82 |
| | Random | 20.21 | 7.25 | 2.64 | 0.62 | 13.89 | 55.16 | 22.07 |
| | Whole | 26.82 | 9.83 | 3.75 | 1.61 | 14.66 | **58.08** | 23.28 |
| | BPC | **34.95** | **14.53** | **5.62** | **1.78** | **23.79** | 49.95 | **32.03** |
| **Sub-2** | None | 26.23 | 9.54 | 3.46 | **1.44** | **50.28** | 17.41 | 25.69 |
| | Random | 21.02 | 7.55 | 2.35 | 0.57 | 11.75 | **62.75** | 19.64 |
| | Whole | 27.26 | 10.19 | 3.14 | 1.06 | 16.52 | 53.10 | 24.90 |
| | BPC | **32.46** | **11.77** | **3.95** | 0.84 | 24.90 | 38.43 | **30.01** |
| **Sub-3** | None | 26.89 | 10.11 | 3.84 | 1.78 | 15.72 | 55.13 | 24.31 |
| | Random | 22.46 | 8.61 | 3.33 | 1.50 | 11.09 | **64.83** | 18.81 |
| | Whole | 29.09 | 11.29 | **4.41** | **1.92** | 18.02 | 49.27 | 26.28 |
| | BPC | **33.22** | **12.91** | 4.29 | 1.76 | **23.22** | 44.31 | **30.24** |

Table 11: The performance of PREDFT when different ROIs are selected for the side network under cross-subject decoding setting in Narratives dataset.

| Length | ROIs | BLEU-1 | BLEU-2 | BLEU-3 | BLEU-4 | ROUGE1-R | ROUGE1-P | ROUGE1-F |
|--------|------|--------|--------|--------|--------|----------|----------|----------|
| 10 | None | 18.08 | 3.98 | 1.05 | 0.28 | 14.96 | 26.21 | 18.96 |
| | Random | 16.83 | 3.04 | 0.63 | 0.13 | 16.71 | 19.23 | 17.28 |
| | Whole | 21.99 | 4.51 | 0.83 | 0.25 | **17.36** | 22.83 | 19.43 |
| | BPC | **24.73** | **8.39** | **3.92** | **1.86** | 14.07 | **35.28** | **19.53** |
| 20 | None | 20.37 | 3.86 | 1.03 | 0.19 | 17.42 | 22.15 | 19.45 |
| | Random | 16.11 | 3.28 | 0.55 | 0.12 | 19.27 | 24.29 | 21.44 |
| | Whole | 23.55 | **6.39** | 1.33 | **0.39** | 15.66 | **30.98** | 20.72 |
| | BPC | **25.98** | 5.61 | **1.36** | 0.21 | **19.61** | 25.43 | **22.09** |
| 40 | None | 18.01 | 4.72 | 1.27 | 0.34 | 16.41 | 34.36 | 22.16 |
| | Random | 19.71 | 5.01 | 1.22 | 0.39 | 20.02 | 29.61 | 24.55 |
| | Whole | 24.67 | 5.81 | 1.14 | 0.39 | **20.53** | 29.46 | 24.16 |
| | BPC | **27.80** | **8.29** | **2.00** | **0.54** | 19.53 | **38.95** | 25.96 |

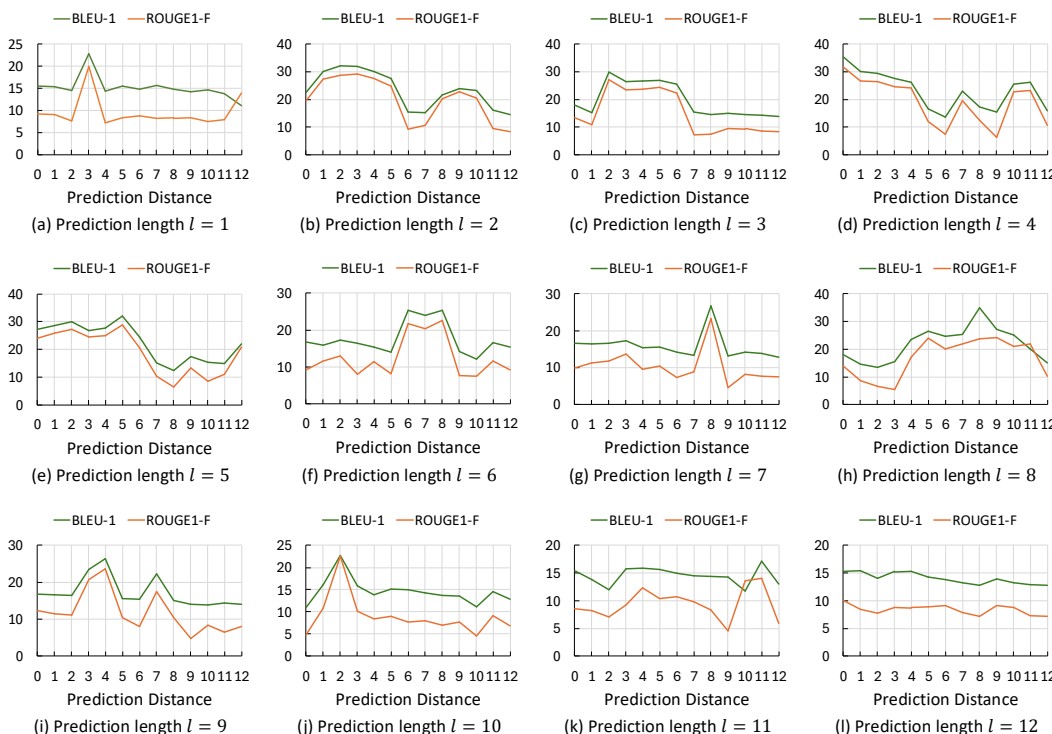

Figure 17: The impact of prediction length and prediction distance on decoding performance of subject-1 in LeBel's dataset.

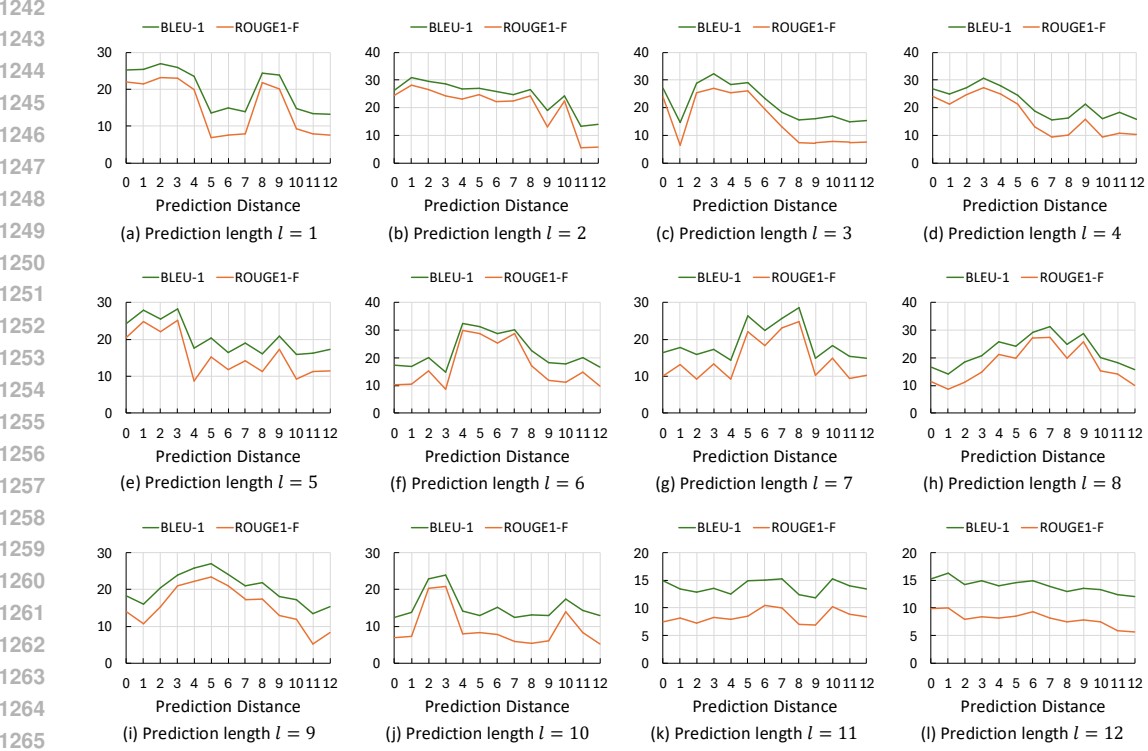

Figure 18: The impact of prediction length and prediction distance on decoding performance of subject-2 in LeBel's dataset.

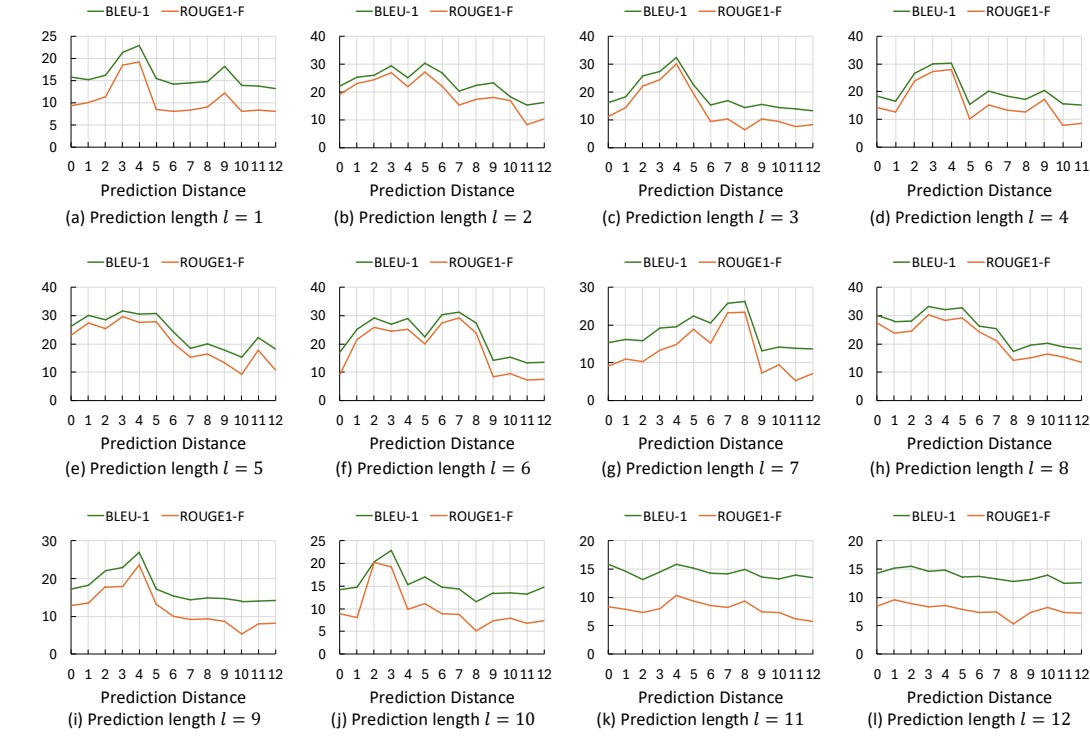

Figure 19: The impact of prediction length and prediction distance on decoding performance of subject-3 in LeBel's dataset.

