# OpenReview forum: "Language Reconstruction with Brain Predictive Coding from fMRI Data"
_ICLR.cc/2025/Conference — Submitted to ICLR 2025_

### Official Review · Reviewer_EgDV · 2024-10-30

**Soundness:** 3
**Presentation:** 2
**Contribution:** 2
**Rating:** 5
**Confidence:** 4

**Summary:**

Recent brain decoding studies have demonstrated that speech perception can be decoded from fMRI recordings and subsequently reconstructed as continuous language. These studies reconstruct continuous language either from specific regions of interest (ROIs) or from the whole brain, using decoder-based language models like GPT-2. Additionally, recent predictive coding studies reveal that the human brain naturally engages in continuously predicting future words across multiple timescales. Building on recent linguistic brain decoding research and the predictive coding approach, this paper explores predictive coding theory in the context of continuous language reconstruction. To this end, the authors propose PREDFT (fMRI-to-Text decoding with Predictive Coding), which consists of a main decoding network and a side network (the predictive coding component). Experimental results on two naturalistic brain datasets (Moth Radio Hour and Narratives) indicate that PREDFT achieves superior decoding performance when comparing the actual story with the reconstructed story.

**Strengths:**

1. The motivation for using predictive coding in continuous language reconstruction is clear and well-explained.
2. The proposed approach aims to improve the reconstruction of narrative stories from fMRI brain data. This is a very interesting research area because reconstructing language is challenging due to the slowness of the hemodynamic response.
3. The authors compared the reconstruction performance using evaluation metrics against recent studies. Additionally, ablation studies were conducted on the proposed approach, with and without the predictive coding component.

**Weaknesses:**

1. There are several major weaknesses in this work, particularly concerning the evaluation of reconstruction results:
	- A major concern is that the current study (PREDFT) does not provide a clear evaluation of reconstruction results compared to the baseline paper by Tang et al. (2023).
	- For example, the authors did not evaluate the word rate in the generated narrative story. Since the fMRI data was captured while participants were listening to stories, each word has an onset and offset. Similarly, during decoding, what is the word rate predicted by the proposed model, and does this word rate match the actual word rate of the original stimuli?
	- Therefore, comparing the reconstructed stimulus to the ground truth (i.e., the actual transcripts of the stimuli) would provide a good sense of whether the outputs are meaningful, as the dataset includes the ground truth of what words participants heard and when they heard them.

2. Furthermore, the authors performed decoding using either random selections of ROIs, the whole brain, or BPC, which includes language-related ROIs. However, prior studies have focused on specific ROIs, such as the language, prefrontal, and auditory association cortices. Therefore, it is unclear how the proposed method compares with prior methods. Since the authors' main research question revolves around how semantic information is embedded in brain signals to improve decoding, they should consider these ROIs, as they maintain a hierarchy of language processing.
	- The random selection of ROIs generally leads to low decoding performance. What are these random ROIs? Do they have any overlap with BPC ROIs?
	- Previous studies have conducted both quantitative and qualitative analyses, reporting what the stimulus decoded at each ROI, including language-related regions in both the left and right hemispheres, as well as using four evaluation metrics. However, this paper does not report any reconstructed stimulus in the main content, nor does it include analysis at the ROI level. Additionally, the authors only used two metrics, and throughout the paper, the focus is more on the scores rather than on the main reconstructed language results.

3. Although the authors report some results on predictive length and distance from the current word in Figure 1, there are no qualitative reconstruction results for these different predictive lengths and distances. What type of information is the model forecasting based on brain data? Is it syntactic information, such as nouns and verbs, or semantic content? This analysis is clearly missing from the paper.

4. All the figures lack detailed captions. The results presented in the figures are difficult to understand. For instance, what is the prediction score in each subplot of Figure 1? What does each line in the top plots represent? What does prediction distance "d" refer to? Without providing clear details in the figure captions or placing the figures appropriately in the text, it becomes challenging for readers to understand the content and what is being conveyed.

5. Since the authors use two encoders and two decoders in the proposed PREDFT, it is unclear which component is primarily responsible for reconstructing the language and which component provides the theme and narrative structure. It would be interesting if the authors reported the generated stimulus from individual components and from PREDFT as a whole, along with the performance metrics. This would help identify the shared and individual contributions of each component during language reconstruction.

**Questions:**

1. What would be the chance-level performance when reconstructing continuous language? Is there a baseline available for comparison? Additionally, what is the percentage of overlap between random ROIs and whole-brain voxels? Did the authors repeat the selection of random ROIs multiple times to ensure robustness, or did they only select a single set of random ROIs?
2. What is the rationale for using 4D volume data from the Narratives dataset while using 2D brain data from the Moth Radio Hour dataset? Since the Narratives dataset includes both smoothed and unsmoothed versions, along with brain masks to select activated voxels from the 4D volume, why did the authors make these choices regarding data representation?
3. There is no interpretation provided for the two encoders used in PREDFT. The authors could project these voxels onto brain maps to verify the quality of their encoders.
4. Figures 3, 4, 6, and 8 appear redundant. The authors could combine these into a single figure with a comprehensive caption, instead of presenting multiple, repetitive figures.
5. What does the y-axis represent in Figure 9?
5. Several major questions are raised in the weaknesses section.

Typos:

1. Line 35: Bhattasali et al. (2019); Wang et al. (2020); Affolter et al. (2020); Zouet al. (2021)  - > (Bhattasali et al. 2019; Wang et al. 2020; Affolter et al. 2020; Zouet al. 2021)

---

> ### Author Response · Authors · 2024-11-15
> **Rebuttal (part 1)**
>
> We sincerely appreciate your effort in reviewing our paper. We will address your concerns point by point.
>
> ### Weaknesses
>
> #### Weakness 1
>
> > There are several major weaknesses in this work, particularly concerning the evaluation of reconstruction results. A major concern is that the current study (PREDFT) does not provide a clear evaluation of reconstruction results.
>
> **We have already provided a case analysis containing reconstruction results in Appendix F.** If you are still confused about the reconstruction results, please refer to our reply to reviewer XUt4.
>
> > For example, the authors did not evaluate the word rate in the generated narrative story.
>
> There's no word rate model in PredFT. We don't use a fixed LLM to endlessly generate decoded content like Tang's model, in which case a word rate model is needed to judge the end point. Instead, we view fMRI-to-text decoding as sequence-to-sequence translation task and train PredFT in an end2end manner.
> So how many words should be decoded is already learned by PredFT. We only need to set a sampling method (e.g. greedy decoding).
>
> #### Weakness 2
>
> > However, prior studies have focused on specific ROIs, such as the language, prefrontal, and auditory association cortices...
>
> In the main decoding network, we apply whole brain data for reconstruction instead of selecting specific ROIs related to language comprehension. In the side network, we use BPC area for language reconstruction, which covers most part of language related areas like auditory cortex (AC), prefrontal cortex (PFC) and Broca area.
>
> > The random selection of ROIs generally leads to low decoding performance. What are these random ROIs? Do they have any overlap with BPC ROIs?
>
> The specific ROIs we applied are listed in Appendix A.4. For example, for narratives dataset G_and_S_cingul-Ant, G_and_S_subcentral, G_and_S_transv_frontopol, G_orbital, S_front_middle, S_subparietal are selected. For LeBel's dataset, we randomly choose 1000 voxels from brain surface data besides BPC area(because fMRI in this dataset isn't projected to a standard space (e.g. MNI space), so we can't apply a brain atlas for parcellation). There is no overlap between Random ROIs and BPC ROIs.
>
> > this paper does not report any reconstructed stimulus in the main content, nor does it include analysis at the ROI level.
>
> The reconstructed stimulus is shown in Appendix F. Since we use whole brain data in main decoding network for language reconstruction, there's no available ROI level reconstructed stimulus.
> We also notice that only Tang's work conducted ROI level reconstructed stimulus, while other previous works (BrainLLM, unicorn, mapguide, BP-GPT) didn't include it.
>
> > Additionally, the authors only used two metrics, and throughout the paper.
>
> We add another metric BERTScore. Please refer to rebuttal part 2.
>
> #### Weakness 3
>
> > there are no qualitative reconstruction results for these different predictive lengths and distances.
>
> When the prediction length and distance is too long or short, it becomes distraction and the decoded content becomes meaningless. No essential concepts or semantics can be decoded. So we think there's no need to show them. The reconstruction results with proper length and distance are shown in Appendix F.
>
> > What type of information is the model forecasting based on brain data?
>
> The training objective for side network is cross-entropy loss between generated words and label, and the label is ground truth word sequence with certain prediction length and distance from the onset word of each fMRI frame. So the prediction is supposed to be the combination of syntactic and semantic information. We can't show what exactly the model predicts since the predictive information is presented in the form of representation, which is a deep learning approach. We can only judge whether it works through reconstruction performance (e.g. bleu).
>
> #### Weakness 4
>
> > All the figures lack detailed captions.
>
> The explanation for concepts like "prediction score", "prediction distance" is presented in the first paragraph of section 2. We will add more captions in the edited version of paper for better understanding.
>
> #### Weakness 5
>
> > it is unclear which component is primarily responsible for reconstructing the language and which component provides the theme and narrative structure.
>
> The main decoding network in PredFT is responsible for whole text reconstruction, while the side network provides predictive coding information and the decoder in side network is discarded after training. **Therefore, no specific part is claimed to be responsible for reconstructing the language, nor is any particular part claimed to provide the theme and narrative structure, the decoder in main decoding network generates as a whole.**

---

> ### Author Response · Authors · 2024-11-15
> **Rebuttal (part 2)**
>
> ### Questions
>
> #### Q1
>
> > What would be the chance-level performance when reconstructing continuous language?
>
> We are conducting a chance-level baseline, please lend us more time.
>
> > what is the percentage of overlap between random ROIs and whole-brain voxels?
>
> None overlapping.
>
> > Did the authors repeat the selection of random ROIs multiple times to ensure robustness?
>
> Yes, the selection of random ROIs is repeated five times and the results are similarly poor.
>
> #### Q2
>
> > What is the rationale for using 4D volume data from the Narratives dataset while using 2D brain data from the Moth Radio Hour dataset?
>
> We selected 4D volumetric whole-brain data from the Narratives dataset to align with the settings used in Unicorn, allowing us to compare performance of different models.
>
> #### Q3
>
> > There is no interpretation provided for the two encoders used in PREDFT.
>
> Thanks for raising this problem. We think directly evaluating the quality of encoder representation is hard, since we apply a deep learning approach instead of a linear encoding model.
> Using the change of decoding performance to judge the quality of encoder representation is an alternative and meaningful approach.
> Project the representation to brain map doesn’t seem to lead to results with practical meaning. Previous works didn’t test the encoding quality in this way either.
>
> #### Q4
>
> > Figures 3, 4, 6, and 8 appear redundant.
>
> We hope showing different components separately can help readers understand the model easily. We also provide the whole framework of PredFT in Figure 11 in the Appendix.
>
> #### Q5
>
> > What does the y-axis represent in Figure 9?
>
> Y axis represents the numerical value of score in a percentage-based system (e.g. 40 means 40%). We will polish the figure.
>
> #### Typos
>
> Thanks for pointing out. We will fix it.
>
> ### BERTScore supplementary
>
> |    |                    |       |
> |----|--------------------|-------|
> |  | Tang's             | 80.84 |
> |    | BrainLLM           | **83.26** |
> |   S1 | MapGuide           | 82.66 |
> |    | PredFT w/o SideNet | 81.35 |
> |    | PredFT             | 82.92 |
> |    |                    |       |
> |  | Tang's             | 81.33 |
> |    | BrainLLM           | **83.4**  |
> |  S2  | MapGuide           | 82.78 |
> |    | PredFT w/o SideNet | 81.42 |
> |    | PredFT             | 82.52 |
> |    |                    |       |
> |  | Tang's             | 81.5  |
> |    | BrainLLM           | **83.82** |
> |  S3  | MapGuide           | 82.84 |
> |    | PredFT w/o SideNet | 81.48 |
> |    | PredFT             | 82.11 |
> ||||
>
> |    |                    |       |
> |----|--------------------|-------|
> | | unicorn            | 75.35 |
> | 10   | PredFT w/o SideNet | 75.26 |
> |    | PredFT             | **78.52** |
> |    |                    |       |
> | | unicorn            | 74.88 |
> |  20  | PredFT w/o SideNet | 75.16 |
> |    | PredFT             | **78.2**  |
> |    |                    |       |
> | | unicorn            | 74.4  |
> |  40  | PredFT w/o SideNet | 75.07 |
> |    | PredFT             | **78.63** |
> |    |                    |       |

---

> > ### Comment · Reviewer_EgDV · 2024-11-22
> >
> > Thank you, authors, for providing clarifications on several points, such as the overlap of ROIs. However, I remain unconvinced by some of the responses and new results, and several questions still need to be addressed.
> >
> > * As someone familiar with brain decoding works and reconstruction processes, I find the authors' response regarding the inability to show exactly what the model predicts due to the representational nature of the predictive information in deep learning approaches unconvincing. While the authors argue that reconstruction performance (e.g., BLEU scores) is the only feasible measure, the core of language model and brain alignment studies in either encoding or decoding lies in interpreting the model using brain data and explaining brain information processing through language models. The focus should not solely be on reconstruction results but also on understanding and explaining the inner workings that lead to reconstruction. If the authors could address this, it would allow a clearer exploration of the limitations of the current language model and open avenues to investigate other models within the NLP community.
> >
> > * Additionally, I agree with reviewer ZU6W's concern that the BERTScore could be artificially boosted by short, high-frequency words rather than meaningful reconstruction, particularly since the cited value represents the recall-only computation of BERTScore, as favored by Tang et al., rather than the traditional F1-score metric. Furthermore, the BERTScores reported in Rebuttal 2 are not convincing. For the Moth-Radio-Dataset, the BrainLLM model consistently outperforms the current model. If the authors claim that the reconstruction text quality of BrainLLM is not compared with the current method, they should investigate the reasons for BrainLLM’s higher scores despite incorrect reconstruction results.
> >
> > * The authors also have yet to perform chance-level analyses for each subject and dataset. This omission is significant and needs to be addressed.
> >
> > * I also disagree with the response from the reviewer stating that no specific part of the model is claimed to be responsible for language reconstruction or providing the theme and narrative structure, as the decoder generates results holistically. If the authors are using different components in the model and building an end-to-end framework for reconstruction, it is essential to articulate the role of each component. What is the hypothesis for including each component, and what specific aspect does it aim to improve? Clear justification for the inclusion of each component is critical.
> >
> > * The statement that "directly evaluating the quality of encoder representations is difficult due to the use of a deep learning approach rather than a linear encoding model is unconvincing". Whether deep learning or linear encoding is used, the fundamental goal of Neuro-AI studies is interpretation. If the authors are selecting voxels from the brain but do not demonstrate clear control over the quality of the encoders, it raises concerns that reconstruction results may be entirely driven by the deep learning model. This concern must be addressed thoroughly.

---

> > > ### Author Response · Authors · 2024-11-23
> > > **Clarifications of five questions (part1)**
> > >
> > > Thanks for the reviewer's reply. Here we address new concerns point by point.
> > >
> > > > I find the authors' response regarding the inability to show exactly what the model predicts due to the representational nature of the predictive information in deep learning approaches unconvincing...
> > >
> > > We agree with the reviewer on the significance of interpreting how the model learns brain information and achieves language reconstruction. However, we don't think we can directly explain or analyze this process (how brain signals become language inside model) with current deep learning approaches.
> > > Obviously, the mechanism of Transformer-based language model remains unknown, so researchers usually seek to find a proper indicator and analyze model function by observing the change of indicator. Just like previous work on predictive coding in brain encoding research [1], the authors observe the change of brain score to figure out the extent of predictive coding.
> > > In brain-to-text decoding, Tang's paper also analyze the model (e.g. effect of ROIs, etc.) by observing the change of text similarity metrics (e.g. bleu).
> > >
> > > In our paper, we do the same way as previous work and mainly conduct three types of experiment to investigate the PredFT model: (1) ROIs analysis (2) Prediction length analysis (3) Prediction distance analysis. Experimental results highlight that (1) The sidenet successfully models predictive coding in brain decoding as reflected by text similarity metrics (2) Predictive coding can improve decoding just like it can improve brain encoding.
> > > We think these experiments are sufficient and convincing.
> > >
> > > We also appreciate any suggestions from reviewers that could help improve model interpretability.
> > >
> > > [1]. Evidence of a predictive coding hierarchy in the human brain listening to speech
> > >
> > > > BERTScore could be artificially boosted by short, high-frequency words rather than meaningful reconstruction...
> > >
> > > We have to clarify that we apply BERTScore-F1 here instead of BERTScore-Recall in Tang's paper, which should make the results more convincing. Despite BrainLLM beats PredFT in BERTScore, the gap is quite narrow. Moreover, when it comes to BLEU and ROUGE metrics, PredFT outperforms BrainLLM a lot. So PredFT can be said to generally perform better than BrainLLM.
> > >
> > > > Lack of chance-level experiment.
> > >
> > > We add the result of chance-level experiment in part 2. Specifically, we randomly shuffle the order of input fMRI frames while maintaining the same model hyper-parameters.
> > >
> > > > I also disagree with the response from the reviewer stating that no specific part of the model is claimed to be responsible for language reconstruction...
> > >
> > > The general design of different functions in different part of PredFT is clear: (1) The main network encoder is supposed to represent fMRI signals. (2) The main network decoder is supposed to combine fMRI representation and predictive coding representation and generate reconstructed text. (3) The sidenet encoder is supposed to represent predictive coding. (4) The sidenet decoder is designed to facilitate the training of sidenet encoder, and will be discarded when the training finishes.
> > >
> > > End-to-end model shows its superiority in simplicity and powerful potential compared to pipeline model, and has been widely adopted in deep learning.
> > > However, one of its drawbacks is the lack of interpretability.
> > > Thus we can't describe a more detailed function of each part, and the absence of such analysis doesn't necessarily have to be seen as a drawback.
> > >
> > > > The statement that "directly evaluating the quality of encoder representations is difficult due to the use of a deep learning approach rather than a linear encoding model is unconvincing"...
> > >
> > > The quality of predictive coding representation is reflected through the change of text reconstruction quality. We can't directly evaluate the predictive coding representation. Generally speaking, there're several approaches to evaluation representation in representation learning: (1) downstream tasks (e.g. image classification in cv) (2) clustering. While in this brain-related scenario all these methods don't work. We consider it to be a promising topic for future research, yet currently we have no ideas regarding it.
> > >
> > > We also appreciate any suggestions from reviewers that could help evaluate predictive coding representation directly.
> > >
> > > **In conclusion, the reviewer is concerned about the interpretability of PredFT in some aspects. We aim to argue that it is impossible to analyze every single aspect of a deep learning model (a well-known fact), and we have already exerted efforts to analyze the parts that are within our reach.**
> > >
> > > Thank you once again for your time and effort in reviewing our submission. We sincerely hope you may reconsider your rating if you find our rebuttal helpful.

---

> > > ### Author Response · Authors · 2024-11-23
> > > **Clarifications of five questions (part2)**
> > >
> > > | LeBel's dataset    |                     | BLEU1 | BLEU2 | BLEU3 | BLEU4 | ROUGE-R | ROUGE-P | ROUGE-F | BERTScore-F1 |
> > > |--------------------|---------------------|-------|-------|-------|-------|---------|---------|---------|--------------|
> > > | sub-1              | PredFT              | 34.95 | 14.53 | 5.62  | 1.78  | 23.79   | 49.95   | 32.03   | 82.92        |
> > > |                    | Chance-level PredFT | 20.34 | 3.75  | 0.2   | 0     | 15.48   | 20.41   | 17.45   | 77.7         |
> > > | sub-2              | PredFT              | 32.46 | 11.77 | 3.95  | 0.84  | 24.9    | 38.43   | 30.01   | 82.52        |
> > > |                    | Chance-level PredFT | 18.96 | 2.96  | 0     | 0     | 15.04   | 20.37   | 17.18   | 78.02        |
> > > | sub-3              | PredFT              | 33.22 | 12.91 | 4.29  | 1.76  | 23.22   | 44.31   | 30.24   | 82.11        |
> > > |                    | Chance-level PredFT | 19.48 | 3.58  | 0.28  | 0     | 15.17   | 19.35   | 16.96   | 78.24        |
> > > |                    |                     |       |       |       |       |         |         |         |              |
> > > |                    |                     |       |       |       |       |         |         |         |              |
> > > | Narratives dataset |                     | BLEU1 | BLEU2 | BLEU3 | BLEU4 | ROUGE-R | ROUGE-P | ROUGE-F | BERTScore-F1 |
> > > | 10                 | PredFT              | 24.73 | 8.39  | 3.92  | 1.86  | 14.07   | 35.28   | 19.53   | 78.52        |
> > > |                    | Chance-level PredFT | 19.92 | 2.6   | 0     | 0     | 13.28   | 22.8    | 16.72   | 76.23        |
> > > | 20                 | PredFT              | 25.98 | 5.61  | 1.36  | 0.21  | 19.61   | 25.43   | 22.09   | 78.2         |
> > > |                    | Chance-level PredFT | 19.53 | 2.45  | 0     | 0     | 14.94   | 21.55   | 17.58   | 75.39        |
> > > | 40                 | PredFT              | 27.8  | 8.29  | 2     | 0.54  | 19.53   | 38.95   | 25.96   | 78.63        |
> > > |                    | Chance-level PredFT | 20.31 | 2.88  | 0.41  | 0     | 15.39   | 24.76   | 18.8    | 75.58        |

---

### Official Review · Reviewer_my4S · 2024-10-31

**Soundness:** 3
**Presentation:** 3
**Contribution:** 3
**Rating:** 5
**Confidence:** 5

**Summary:**

In the submission-6263, the authors  propose PREDFT (FMRI-to-Text decoding with Predictive coding) , which was inspired by predictive coding theory. This theory suggests that when humans listen to a certain speech, their subconscious brain predicts the words they may hear next. Then the author validated this theory through a prediction score. The verification method is to first calculate the correlation coefficient between the features extracted by LLM at the current location and the brain features, and then add the features of an upcoming text segment to the current location features, calculate the correlation coefficient again, and observe the changes in the correlation coefficient. The experimental results show that incorporating upcoming text features can increase the correlation coefficient between LLM features and brain features. Based on the above experimental results, the author designed their own model, which includes the side network to decode upcoming text. In the decoding of current text, the feature from the side network is used to incorporate the predictive coding theory into the method.

**Strengths:**

The author provided sufficient experiments to demonstrate the significance of his motivation.

**Weaknesses:**

Although the author's explanation of motivation is very sufficient, I still have a few major questions about the author's method and list them in the questions part.

**Questions:**

(1) Why did the author only use BLEU and ROUGE in the experiment? Why doesn't the author use WER, METEOR, and BERTScore which is used in the Tang and MapGuide? BLEU and ROUGE both evaluate the matching degree of n-grams, which can easily lead to surface matching but semantic mismatch. METEOR and BERTScore can better reflect semantic similarity.
(2) Many of the methods compared by the author incorporate LLM, while the author's model is entirely trained with their own transformer. Does this result in the author's method being inferior to the baseline method in terms of semantic similarity?
(3) The author's method was inspired by predictive coding and validated it on LLM using a prediction score. But can the author's own model still observe the same phenomenon on the prediction score? I haven't seen the same experiment evaluating the author's own model.
(4) In some parts of the paper, fMRI is spell as FMRI.

---

> ### Author Response · Authors · 2024-11-14
> **Rebuttal**
>
> We sincerely appreciate your effort in reviewing our paper. We will address your concerns point by point.
>
> > Why did the author only use BLEU and ROUGE in the experiment?
>
> We choose BLEU and ROUGE in the experiment to follow the evaluation setting in UniCoRN [1]. Generally speaking, BLEU, ROUGE, WER, METEOR all measure word-level overlapping between generated content and ground truth. Choosing any of them will lead to similar results, while BERTScore is designed to evaluate semantic-level similarity. **We add the BERTScore result of different models here for supplementary.**
>
> |    |                    |       |
> |----|--------------------|-------|
> |  | Tang's             | 80.84 |
> |    | BrainLLM           | **83.26** |
> |   S1 | MapGuide           | 82.66 |
> |    | PredFT w/o SideNet | 81.35 |
> |    | PredFT             | 82.92 |
> |    |                    |       |
> |  | Tang's             | 81.33 |
> |    | BrainLLM           | **83.4**  |
> |  S2  | MapGuide           | 82.78 |
> |    | PredFT w/o SideNet | 81.42 |
> |    | PredFT             | 82.52 |
> |    |                    |       |
> |  | Tang's             | 81.5  |
> |    | BrainLLM           | **83.82** |
> |  S3  | MapGuide           | 82.84 |
> |    | PredFT w/o SideNet | 81.48 |
> |    | PredFT             | 82.11 |
> ||||
>
> |    |                    |       |
> |----|--------------------|-------|
> | | unicorn            | 75.35 |
> | 10   | PredFT w/o SideNet | 75.26 |
> |    | PredFT             | **78.52** |
> |    |                    |       |
> | | unicorn            | 74.88 |
> |  20  | PredFT w/o SideNet | 75.16 |
> |    | PredFT             | **78.2**  |
> |    |                    |       |
> | | unicorn            | 74.4  |
> |  40  | PredFT w/o SideNet | 75.07 |
> |    | PredFT             | **78.63** |
> |    |                    |       |
>
>
> Results show our model performs good in semantic-level reconstruction (the gap compared to the best is narrow), while maintaining the best BLEU and ROUGE score.
>
> [1]UniCoRN: Unified Cognitive Signal ReconstructioN bridging cognitive signals and human language. ACL'23
>
> > Many of the methods compared by the author incorporate LLM, while the author's model is entirely trained with their own transformer. Does this result in the author's method being inferior to the baseline method in terms of semantic similarity?
>
> The above supplementary BERTScore result confirms that our model achieves similar semantic reconstruction ability as LLM-based method. Moreover, the disadvantage of LLM-based method is that the parameters of LLM is fixed and can't be finetuned. Only the input of LLM could be changed and previous methods (Tang's model, BrainLLM, MapGuide) try different ways to improve input representation. While training Transformer based model from scratch is more flexible (allowing architecture changes).
>
> > The author's method was inspired by predictive coding and validated it on LLM using a prediction score. But can the author's own model still observe the same phenomenon on the prediction score? I haven't seen the same experiment evaluating the author's own model.
>
> We have to clarify that they're two completely different types of tasks:
>
> - Prediction score
>
>   It belongs to brain-LM alignment research (also termed brain encoding in some paper). The common method is mapping (e.g. via linear model, rsa) the output of LM layer to brain response, and analyzing voxel-level correlation. It aims to discover whether LM has human-like language comprehension ability.
>
> - Language reconstruction
>
>   The task is also known as brain decoding, fMRI-to-text decoding. It aims to decode natural language from brain recordings.
>
> As a result, it's meaningless to calculate the prediction score for PredFT because it's not a language model.
> Even if we do so, the output result lacks practical meaning.
> We present the prediction score experiment to highlight how we get the motivation of PredFT: while predictive coding improves brain encoding, could it help improve brain decoding?
>
> > In some parts of the paper, fMRI is spell as FMRI.
>
> fMRI is written as FMRI only in Abstract to highlight how we name the model: the **FT** in PredFT comes of **F**MRI-to-**T**ext.

---

> > ### Comment · Reviewer_my4S · 2024-11-15
> > **The experimental results cannot achieve higher rating.**
> >
> > While this article presents strong experimental results that highlight the significance of its motivation, its model performance falls short of the state-of-the-art (SOTA) in several metrics. Specifically, this is evident in BLEU-3, BLEU-4, ROUGE-R, and ROUGE-P scores for certain subjects, as well as the BERTScore across all subjects on LeBel's dataset. These results suggest that there remains room for improvement if the authors aim to leverage Predictive Coding theory to enhance fMRI decoding methods. This performance gap is also a primary reason for my lower rating.
> >
> > Regarding the use of predictive scores to evaluate the proposed method, I believe the output layer of the decoder in the authors’ model plays a role analogous to the output layer of a language model (LM). Therefore, it could reasonably be treated as an LM for the purpose of calculating predictive scores. Experiments utilizing predictive scores would help demonstrate whether the decoder’s output layer in the main and future networks exhibits the same predictive coding phenomenon as the corresponding fMRI signals. Furthermore, the author provide results on the impact of predictive distance on metrics in the appendix made me curious. Specifically, when predictive distance changes, does the predictive score（or an alternative similarity metric, if the authors find predictive scores inappropriate） between the decoder's features and fMRI signals align with changes in the metrics? I encourage the authors to consider this experiment and provide explanations for the results, as this would help clarify whether predictive coding improves fMRI decoding performance.
> >
> > The authors have their perspective on the use of predictive scores, arguing that such scores must be calculated between an LM and brain responses, and their model is a neural decoding model rather than an LM. However, given the performance issue of their method, I view the question of predictive scores as a secondary concern.
> >
> > Finally, since the authors have presented results showing the impact of predictive distance on the metrics, it is reasonable for reviewers to question the underlying causes of these effects. This naturally draws attention to the predictive score mentioned in the paper. If the authors do not plan to include additional experiments to explore this issue in future work, it is recommended that they provide a detailed explanation within the paper.

---

> > > ### Author Response · Authors · 2024-11-15
> > > **Thanks for your reply, and we would like to highlight the contribution of this work.**
> > >
> > > Thanks for the review's reply. We find your primary concern lies in that our model fails to outperform previous methods in a very small part of evaluation metrics.
> > > Model performance is crucial without doubt. **But we don't agree that a model needs to beat previous methods in every single aspect to be called a "good" one.**
> > > We can list many top AI conference papers that do not outperform previous methods in every metric (a simple case is a better LLM A doesn't need to beat LLM B in every benchmark).
> > > More importantly, besides the improvement of decoding performance, the most significant contribution of our paper is that we first design a model that combines neuroscience theory and deep learning in fMRI-to-text decoding, while previous methods focused on pure deep learning tricks. It has great potential to inspire future work in this domain.
> > >
> > > As to your secondary concern, **we believe the best way to clarify whether predictive coding improves fMRI decoding performance is to evaluate the decoding performance itself, but not to observe the correlation between some layer of PredFT and brain response (i.e. prediction score).** We designed several ablation studies to show the effectiveness of predictive coding: PredFT w/o predictive coding, ROIs selection, prediction length and distance study. In short, we have addressed this concern in our paper.
> > >
> > > As to your final concern, we will add relative discussion in edited version of paper. Here's our explanation: We think it's the model design of PredFT that leads to this phenomenon: We apply a Transformer based SideNet for modeling predictive coding, so too long and far predicted content will lead to poor training because of the noise introduced. A short and close predicted information will lead to better predictive coding representation, thus improving decoding performance. This modeling is similar to the actual predictive coding mechanism in human brain: it only makes prediction about near future.
> > >
> > > We sincerely hope the reviewer may reconsider the rating given our clarification.

---

> > > > ### Comment · Reviewer_my4S · 2024-11-29
> > > > **The model performance is not convincing. Unable to provide a direct experimental explanation for performance improvement.**
> > > >
> > > > ## 1. Model Performance
> > > >
> > > > I agree with your opinion that a model doesn’t need to beat previous methods in every single aspect to be called a "good" one. And that's also why I asked you to add experiments for METEOR and BERTScore. If I believe that a good model must achieve SOTA on all metrics, then your model's performance on BLEU-3, BLEU-4, ROUGE-R, and ROUGE-P can already demonstrate its performance. **Please do not misinterpret my comment.**
> > > >
> > > > Your comparison method evaluated the semantic consistency of the decoding, but the original version of your experimental results did not include similar metrics. **I believe that you understand the purpose of each metric, so you should also recognize that my request to include two additional indicators was not about determining whether your model achieves SOTA across all metrics. Instead, it was aimed at evaluating the semantic consistency of your reconstruction results. Therefore, the result on BERTScore is not merely “one single aspect”.** The addition experiment of the BERTScore on LeBel's dataset confirmed my concerns, which directly resulted in no higher scores for your work.
> > > >
> > > > ## 2. Unable to provide a direct experimental explanation for performance improvement
> > > >
> > > > I believe the best way to verify whether the motivation behind a method is useful for performance is to examine the consistency between the motivation and the performance outcomes.  **The performance improvement observed after adding the SideNetwork can be attributed to many factors, and you need experiments to demonstrate that this improvement is indeed driven by your stated motivation.**
> > > >
> > > > For your work, the most direct way is to demonstrate that the fusion of features from the SideNetwork and the main network achieves a higher correlation (i.e., prediction score) with brain signals than using only the feature from the main network.  Further, this higher correlation will bring better performance, reflected in the consistency between correlation and performance. **Moreover, if the changes in correlation for these features do not align with the changes in your model's performance, then your explanation for SideNetwork's role is incorrect, and mentioning Section 2 in your paper would be unnecessary.** For example,  when the correlation advantage between fused features and brain signals (over the correlation between main network features and brain signals) differs from the trend shown in Fig 13. In such a case, **you would need to reconsider how the SideNetwork actually contributes to your model.**
> > > >
> > > > Of course, you could opt for other indirect experiments to address this issue, such as exploring the relationship between your model's performance and prediction distance and showing its trend is similar to the trend in Section 2. **However, the persuasiveness of such an approach would be relatively limited compared to the direct way.**

---

> > > ### Author Response · Authors · 2024-11-22
> > >
> > > Dear Reviewer my4S,
> > >
> > > We wanted to kindly follow up again regarding our responses to your comments and feedback on our submission. We have thoroughly addressed the concerns you raised and provided additional analyses in our rebuttal.
> > > If you find our responses satisfactory, we sincerely hope you may reconsider the rating.
> > >
> > > Thank you again for your time and effort in reviewing our work.
> > >
> > > Best regards
> > >
> > > Authors

---

### Official Review · Reviewer_XUt4 · 2024-11-02

**Soundness:** 3
**Presentation:** 3
**Contribution:** 2
**Rating:** 6
**Confidence:** 4

**Summary:**

The paper presents PREDFT (FMRI-to-Text Decoding with Predictive Coding), a novel framework that utilizes predictive coding to translate fMRI signals into continuous language. This approach combines a primary decoding network with an auxiliary network focused on capturing brain predictive coding, aiming to improve the accuracy of language reconstruction from brain signals. The authors conduct experiments on two established naturalistic language comprehension fMRI datasets, showing that PREDFT achieves state-of-the-art performance across multiple evaluation metrics.

**Strengths:**

1. Integrating predictive coding theory into the decoding process offers a fresh perspective on reconstructing language from brain signals.
2. Experimental results demonstrate that PREDFT outperforms other methods across various evaluation metrics, showing significant improvements.

**Weaknesses:**

1. In Section 3.3, the authors state, 'During the inference stage, as illustrated in Figure 8, the decoder in the side network is abandoned.' However, they do not provide a detailed explanation of why the decoder is discarded or discuss the potential impact of this decision. It is recommended to elaborate on the rationale behind this choice and its implications on the overall performance and functionality of the model.

2. As shown in Table 1, PREDFT does not achieve the best performance on ROUGE1-R. The authors should analyze the potential reasons for this and discuss any factors that may have contributed to the lower performance in this specific model. For instance, the model's architecture, training process, or characteristics of the ROUGE1-R metric that might explain the discrepancy. If the authors observed any patterns in the types of language constructs where PREDFT underperformed on ROUGE1-R.

3. As shown in Table 1, PREDFT without SideNet performs similarly to other methods. However, the inclusion of SideNet leads to a significant performance improvement. The authors should provide a detailed analysis of this phenomenon to explain how SideNet contributes to the model's enhanced performance.

4. Although the authors provide a detailed description of the hyperparameter selection, they do not explain the rationale behind these choices. How these choices relate to the model's performance or the underlying theory of predictive coding.

5. "In the regions of interests selection experiment, the authors only consider 'random,' 'whole,' and 'BPC' as the ROIs, which appears somewhat limited. The paper does not clarify whether there are other potential ROIs associated with predictive coding, nor does it provide supporting neuroscience literature for the selection of BPC. It is recommended to either justify the choice of BPC with relevant references or explore additional ROIs to strengthen the study's validity. If authors can explain the process for selecting these particular ROIs and why authors believe these are sufficient to demonstrate the effectiveness of their approach. Additionally, if authors considered any other ROIs and why those were not included in the study.

6. It is recommended that the authors provide pseudocode for the method and an analysis of its time complexity to enhance the reproducibility of the article.

7. The results provided by the authors mostly only include the meanvalue. The experimental results should provide the mean, variance, and statistical test results.

8. In the methods section, some symbols are not defined. It is recommended that the authors compile a list of symbols used in the paper in an appendix to help readers understand better.

**Questions:**

1. In Section 3.3, the authors state that the decoder in the side network is abandoned during the inference stage. Could the authors provide a detailed explanation of why the decoder is discarded and discuss the potential impact of this decision on the overall performance and functionality of the model?

2. As shown in Table 1, PREDFT does not achieve the best performance on ROUGE1-R. Could the authors analyze the potential reasons for this and discuss any factors that may have contributed to the lower performance in this specific model? For instance, how might the model's architecture, training process, or characteristics of the ROUGE1-R metric explain this discrepancy? Did the authors observe any patterns in the types of language constructs where PREDFT underperformed on ROUGE1-R?

3. As shown in Table 1, PREDFT without SideNet performs similarly to other methods, while the inclusion of SideNet leads to a significant performance improvement. Could the authors provide a detailed analysis of this phenomenon to explain how SideNet contributes to the model's enhanced performance?

4. Although the authors provide a detailed description of the hyperparameter selection, could they explain the rationale behind these choices? How do these choices relate to the model's performance or the underlying theory of predictive coding?

5. In the regions of interest selection experiment, the authors only consider 'random,' 'whole,' and 'BPC' as the ROIs. Could the authors clarify whether there are other potential ROIs associated with predictive coding? If so, could they provide supporting neuroscience literature for the selection of BPC? Additionally, can the authors explain the process for selecting these particular ROIs and why they believe these are sufficient to demonstrate the effectiveness of their approach? Did the authors consider any other ROIs, and if so, why were those not included in the study?

6. Could the authors provide pseudocode for the method and an analysis of its time complexity to enhance the reproducibility of the article?

7. The results provided by the authors mostly only include the mean value. Could the authors include the variance and statistical test results in the experimental results?

8. In the methods section, some symbols are not defined. Could the authors compile a list of symbols used in the paper in an appendix to help readers understand better?

---

> ### Author Response · Authors · 2024-11-13
> **Rebuttal (part 1)**
>
> We sincerely appreciate your effort in reviewing our paper. We will address your concerns point by point.
>
> ### Weaknesses
>
> > In Section 3.3, the authors state, 'During the inference stage, as illustrated in Figure 8, the decoder in the side network is abandoned.' However, they do not provide a detailed explanation of why the decoder is discarded or discuss the potential impact of this decision...
>
> What we really want is predictive coding representation, which is produced by side network encoder. The side network decoder is designed to help train the side network encoder (we can't find out how to directly obtain predictive coding representation).
> Specifically, during the training process, the label for side network decoder is predicted words instead of complete sentences (as shown in Figure 3). The side network learns mapping between specific areas of brain (BPC area) and predicted words.
> However, the goal of our task is to decode complete sentences. So the side network decoder is useless after training.
>
> **We will add the motivation and reason for discarding the side network decoder in the methodology section in the updated version.**
>
> > As shown in Table 1, PREDFT does not achieve the best performance on ROUGE1-R...
>
> We think the different lengths of generated content might contribute to this factor, since we don't apply a word rate model to control the number of generated words.
> Although PREDFT fails to outperform other models in ROUGE1-R, the gap is narrow.
> Just like recall and precision to f1 score, ROUGE-Recall measures the extent to which a machine-generated content captures the information contained in a reference content, which is a single perspective assessment. ROUGE-Recall and ROUGE-Precision are characterized by a trade-off. When PREDFT gets a relative low ROUGE-R, it gets a high ROUGE-P.
> Instead, ROUGE-F1 is a more comprehensive indicator, combining both ROUGE-P and ROUGE-R. Our model outperforms other models in this metric.
>
> > As shown in Table 1, PREDFT without SideNet performs similarly to other methods. However...
>
> The SideNet is designed to obtain predictive coding representation.
> PREDFT without SideNet can be viewed as traditional deep learning approach which directly applies Transformer to decode text from brain recordings, while PREDFT with SideNet combines deep learning and neuroscience findings (predictive coding). The SideNet provides predictive coding representation to the decoder in Main network, and the decoder incorporates both current fMRI representation and predictive coding representation for text decoding.
>
> The idea of PREDFT is motivated by predictive coding theory in Neuroscience, which indicates human can naturally predict upcoming words. Since predictive coding has been verified to contribute to human language comprehension, we seek to investigate whether such predictive information can help language reconstruction.
> The improvement of incorporating SideNet highlights 1. the effectiveness of our model design 2. predictive coding has potential to improve brain-to-text decoding. **We will provide the illustration of PREDFT without SideNet for better understanding in the updated version.**
>
> > Although the authors provide a detailed description of the hyperparameter selection...
>
> All the hyperparameters are chosen to minimize the training & validation loss as much as possible.
> We don't understand which hyperparameter the reviewer is confused about.
> The influences of ROIs selection, prediction length, prediction distance,  $ \lambda $ to model performance are detailedly discussed in sec 4.3, sec 4.4, Appendix E. The learning rate is set to stabilize training. We don't test the influence of model layers (e.g. Transformer layers) due to limited computational resources.

---

> ### Author Response · Authors · 2024-11-13
> **Rebuttal (part 2)**
>
> > In the regions of interests selection experiment, the authors only consider 'random,' 'whole,' and 'BPC' as the ROIs...
>
> The selected BPC area (superior temporal sulcus, angular gyrus, supramarginal gyrus, and opercular, triangular, orbital part of the inferior frontal gyrus) contributes the most to predictive coding, as indicated in some neuroscience studies [1][2].
>
> The process for selecting "BPC": For the Narratives dataset, Destrieux atlas is applied and the above mentioned ROIs are extracted. For LeBel's dataset, since the fMRI signals are not projected to a standardized space, we use the “Auditory” region provided by the authors', containing parietal-temporal-occipital (PTO) area. The BPC area of both datasets cover highly similar area.
>
> The process for selecting "random": For the Narratives dataset, G_and_S_cingul-Ant, G_and_S_subcentral, G_and_S_transv_frontopol, G_orbital, S_front_middle, S_subparieta are selected. For LeBel's dataset, we randomly choose 1000 voxels from brain surface data.
>
> The process for selecting "whole": We use the whole brain surface data as ROIs for both datasets.
>
> We believe selecting random and whole ROIs as controlled experiments is sufficient for demonstrating the effectiveness of using predictive coding to improve decoding performance:
>
> 1. random vs. BPC demonstrates only ROIs related to predictive coding in human language comprehension can improve decoding.
>
> 2. whole vs. BPC not only confirms conclusion in 1, but also shows whole brain surface which contains BPC area still can't contribute to better decoding, because some other brain regions contain too much noise.
>
> 3. none (PREDFT without SideNet) vs. BPC. PREDFT without SideNet is equivalent to not using any ROIs for predictive coding. This comparison shows predictive coding improves decoding accuracy significantly.
>
> **All the above clarifications are included in sec 4.3 and Appendix A.4. We will add more key information in sec 4.3 in the updated version**
>
> [1]. Evidence of a predictive coding hierarchy in the human brain listening to speech. Nature Human Behavior
> [2]. Natural speech reveals the semantic maps that tile human cerebral cortex. Nature
>
> > Could the authors provide pseudocode for the method...
>
> We will provide pseudocode in the appendix for edited version of paper. The discussion of time complexity is already in Appendix A.4 (line 845) of original paper.
>
> > The results provided by the authors mostly only include the mean value...
>
> We guess you indicate the experiment of analyzing the impact of prediction length and distance to model performance (sec 4.4), as we have presented results per subject for other experiments. **The per-subject results of analyzing the impact of prediction length and distance are already presented in Figure 16,17,18 in the appendix of original paper.**
>
> > In the methods section, some symbols are not defined...
>
> **A notation table for symbols is presented in Table 3 in the appendix of original paper.**
>
> ### Questions
>
> Please refer to clarification for weaknesses.

---

### Official Review · Reviewer_ZU6W · 2024-11-03

**Soundness:** 2
**Presentation:** 2
**Contribution:** 2
**Rating:** 3
**Confidence:** 5

**Summary:**

The paper describes a decoding method "PredFT" that uses a main decoding network and a side network to perform decoding from fMRI recordings of subjects listening to stories to text. The side network is responsible for obtaining predictive coding representations from specific brain regions and integrating them into the main network, enhancing language decoding. The authors claim that this integration leverages brain regions known for predictive functions (like the parietal-temporal-occipital areas) to better align brain signal decoding with anticipated semantic content. This is supported by results that have claimed the brain performs predictive coding during language stimulation.

**Strengths:**

The attempt to use hypothesized predictive coding representations to enable better text decoding is interesting.

**Weaknesses:**

My main concern is that the metric does not seem to produce even locally coherent text, which substantially damages the authors' claims that this method is an advancement over prior work, such as Tang et al., which uses an LM to guarantee local coherence. Consider the following example from the case study: "He don’t know my girl you of the eyes but his girl sleep he and he said and he said and the to the and and which I not wrong. But the Guy". Clearly, this has no meaning, and does not even obey basic local grammatical rules (e.g. "and and"). The problem seems to be that the model has merely learned repeat short, high-frequency words like "the", "he" and "and", which improves BLEU/ROGUE score but does not actually move forward towards the goal of better language decoding. I imagine if you just had the model repeatedly and randomly output words sampled from the top 100 most common English words that it would behave fairly similarly. My expectation is that a small percentage of the improvement in BLEU score is genuinely derived from brain signals, with most of the benefit deriving from this output bias. The unreasonably high 5.62 BLEU-3 score when compared to other methods is more of a red flag, because its pretty clear that the model is simply guessing every high frequency trigram in the English language.

 The paper is also quite difficult to read for no reason and pointlessly notational, for example when the self-attention equation is repeated three separate times in only slightly different ways.

**Questions:**

Please see weaknesses. I would need to be convinced that majority of the claimed improvements in the model are not merely from a bias towards outputting high-frequency words, and thereby overfitting the chosen test metrics of BLEU and ROGUE, in order to change my score. Right now, I am fairly convinced that this is the case.

---

> ### Author Response · Authors · 2024-11-14
> **Rebuttal (part 1)**
>
> We sincerely appreciate your effort in reviewing our paper. We will address your concerns point by point.
>
> > Unconvincing reconstruction result.
>
> We fully understand your concern. We noticed similar problems during experiments, and here's our explanation.
>
> fMRI-to-text decoding is an important but extremely difficult task due to 1. noisy fMRI signal with latency 2. mismatch between fMRI sampling frequency (2s) and word rate (0.2s). We haven't found any models that can decode satisfying text. If you detailedly read the cases (not cherry picked cases) in previous work (e.g. page S2 in Tang's paper <https://www.biorxiv.org/content/10.1101/2022.09.29.509744v1.full.pdf> ), **you will find that, although the generated content is locally coherent, it's far from being relevant to the ground truth.** We present the decoding results of the first 10 seconds from Tang's paper here for example:
>
> | Truth                                                                           | S1                                                                                                                            | S2 | S3 |
> |---------------------------------------------------------------------------------|-------------------------------------------------------------------------------------------------------------------------------|----|----|
> | I had no shoes on I was crying i had no wallet but i was ok because i had my cigarettes | she said she was a little stressed out because she wasn't doing anything wrong and had a lot of anxiety issues but i had been | i don't have any friends that have one and i really don't care if you do i have a girlfriend i don't mind at all   |  we got in my car and i was crying i didn't have my purse i don't have any money to pay for gas i  |
>
> S3 is considered as good case by the authors, while it only decodes the concept of "crying" and "purse". Not to mention the content of S1 and S2 is completely irrelevant.
> Our model can decode more concepts as shown in our random picked cases. So the question becomes: **Could it be said that a coherent but completely irrelevant text is preferable to an incoherent text but containing more important concepts in fMRI-to-text decoding task?** We believe using metrics to evaluate different models is fair and reliable. BLEU has several drawbacks as you mentioned, which makes BLEU-1 less convincing. But BLEU-3, 4 is reliable. Because there're not many meaningless and repeated trigrams in ground truth, so high BLEU-3,4 prove that model decodes more important concepts. We believe the 5.62 BLEU-3 just shows superiority of our model.
> **Besides, we supplement BERTScore to evaluate semantic similarity for more convincing results in Rebuttal (part 2).** Results show our model performs good in semantic-level reconstruction (the gap compared to the best is narrow), while maintaining the best BLEU and ROUGE score.
>
> Moreover, we have identified the reasons that lead to this phenomenon. We divide existing models into two categories:
>
> 1. The parameters in LM are fixed (Tang's method, BrainLLM, MapGuide).
>
> - Advantages
>
>     - Can generate coherent text with the power of pre-trained LM.
>
> - Disadvantages
>
>     - This approach largely restricts model architecture. Only the input representation can be changed. The innovation of previous works is that they try various ways to enhance input representation.
>
> 2. The parameters in LM are changed (Unicorn, PredFT)
>
> - Advantages
>
>     - The model design is flexible without restriction. Many innovations might be sparked.
>
> - Disadvantages
>
>     - It's hard to maintain good coherence when finetuning LM or training from scratch due to the very complex relationship between fMRI and text (as mentioned in the beginning).
>
> **Both types of models can decode part of important concepts.**
>
> In conclusion, while we still have a long way to go towards coherent and accurate fMRI-to-text decoding, we hope the above clarification convinces you that our work represents a solid step.
>
> > The paper is also quite difficult to read for no reason and pointlessly notational...
>
> The three attention equations represent different meanings:
>
> - Self-attn: q, k, v all come from the input to self-attention layer. It captures relationship between words.
>
> - ED-attn: q comes from the input to self-attention layer, k and v come from the output from main network encoder. It captures relationship between fMRI and words.
>
> - PC-attn: q comes from the input to self-attention layer, k and v come from the output from side network encoder. It captures relationship between predictive information and words.
>
> We hope readers can have a better understanding of how PredFT operates (especially the input of different attention blocks) with the presented equations. We will only keep the predictive coding attention (PC-attn) equation in the edited version if equations become distractions.

---

> ### Author Response · Authors · 2024-11-14
> **Rebuttal (part 2) BERTScore results**
>
> We add BERTScore of different models here.
>
> For LeBel's dataset:
>
> |    |                    |       |
> |----|--------------------|-------|
> |  | Tang's             | 80.84 |
> |    | BrainLLM           | **83.26** |
> |   S1 | MapGuide           | 82.66 |
> |    | PredFT w/o SideNet | 81.35 |
> |    | PredFT             | 82.92 |
> |    |                    |       |
> |  | Tang's             | 81.33 |
> |    | BrainLLM           | **83.4**  |
> |  S2  | MapGuide           | 82.78 |
> |    | PredFT w/o SideNet | 81.42 |
> |    | PredFT             | 82.52 |
> |    |                    |       |
> |  | Tang's             | 81.5  |
> |    | BrainLLM           | **83.82** |
> |  S3  | MapGuide           | 82.84 |
> |    | PredFT w/o SideNet | 81.48 |
> |    | PredFT             | 82.11 |
> ||||
>
> For Narratives dataset:
>
> |    |                    |       |
> |----|--------------------|-------|
> | | unicorn            | 75.35 |
> | 10   | PredFT w/o SideNet | 75.26 |
> |    | PredFT             | **78.52** |
> |    |                    |       |
> | | unicorn            | 74.88 |
> |  20  | PredFT w/o SideNet | 75.16 |
> |    | PredFT             | **78.2**  |
> |    |                    |       |
> | | unicorn            | 74.4  |
> |  40  | PredFT w/o SideNet | 75.07 |
> |    | PredFT             | **78.63** |
> |    |                    |       |

---

> > ### Comment · Reviewer_ZU6W · 2024-11-18
> >
> > I appreciate the additional analysis however I remain somewhat unconvinced here - I am familiar enough with work in this field to know that it is entirely possible to boost BERTScore simply by using short high-frequency words as opposed to long words, especially since the value you cite is the recall-only computation of BERTScore favored by Tang et al rather than the more traditional F1-score metric. This may be unsatisfying to the authors, but the case study simply does not pass the "smell test" for decoding quality. Even in the counterexample cited by the authors from that study, it is clear that there is a greater and more substantial match than in the case study of this paper. Furthermore, semantic similarity means little when the reference text has no semantic coherence to begin with.

---

> > > ### Author Response · Authors · 2024-11-18
> > >
> > > Thanks for the reviewer's reply. We have to clarify that BERTScore-F1 is applied instead of BERTScore-Recall in our supplementary experiment. Moreover, the reviewer still believes Tang's method has a greater and more substantial match than in the case study of this paper. We present more cases from Tang's paper here, and hope the reviewer could **read these cases carefully instead of a quick view**.
> > >
> > > | Truth                                                                           | S1                                                                                                                            | S2 | S3 |
> > > |---------------------------------------------------------------------------------|-------------------------------------------------------------------------------------------------------------------------------|----|----|
> > > | I had no shoes on I was crying i had no wallet but i was ok because i had my cigarettes | she said she was a little stressed out because she wasn't doing anything wrong and had a lot of anxiety issues but i had been | i don't have any friends that have one and i really don't care if you do i have a girlfriend i don't mind at all   |  we got in my car and i was crying i didn't have my purse i don't have any money to pay for gas I  |
> > > |and i didn't want any part of freedom if i didn't have my cigarettes when you live with someone who has a temper a very bad temper a|watching the news and realized that her mom was the most important thing to me and I was not a fan of this guy she was a nice girl i loved|when we do this it is the only thing i do for a living my entire life i'm always going to do something i|wasn't a very good friend to anyone that i had known since my dad was an alcoholic he was abusive to everyone and it was very|
> > > |very very bad temper you learn to play around that you learn this time i'll play possum and next time i'll|her very much but i couldn't let her go without making it clear i wanted a relationship she would say no to everything and be willing to|love and do i can say it now with absolute conviction that you are doing what i want you to do in life is to|hard on me as well as him to do anything that he said or did i would say a word to him|
> > > |just be real nice or i'll say yes to everything or you make yourself scarce or you run and this was one of the times when you just run and as|deal with anything just as long as it didn't mean something she was afraid of i did it a lot of times but never like this and|go out and enjoy the good times and make sure you're happy with the results you get to see them as they are because you have to have them at|in anger or a threatening way in his case it was always just an excuse to leave and that was why i did it and i think that when|
> > > |i was running i thought this was a great place to jump out because there were big lawns and there were cul de sacs and sometimes he would come after me and drive and yell stuff at me to get back in get back in|now i see how she can make it work with a single phone call to the hospital my first instinct is to run over and kick her out of the room and say no you can|that point the next day you are on the street in a neighborhood with no sidewalk so you can't run off the road to escape the cops who have already started chasing you i didn't say no but i|i finally did i ended up moving to an area with very few houses on the property so the neighbors wouldn't hear my car stop in the driveway and run out and tell me to leave and not|
> > >
> > > We don't intentionally pick bad cases: all the shown cases are selected in order. After reading these cases, could it be said the counterexamples has greater and more substantial match?

---

> > > ### Author Response · Authors · 2024-11-22
> > >
> > > Dear Reviewer ZU6W,
> > >
> > > We are reaching out to follow up on our previous message regarding your valuable feedback and our responses in the rebuttal.
> > > If there are any questions or further points you'd like us to address or clarify, we would be more than happy to provide additional details.
> > >
> > > Thank you once again for your time and effort in reviewing our work.
> > >
> > > Best regards
> > >
> > > Authors

---

### Meta-Review · Area_Chair_PKzG · 2024-12-13

**Metareview:**

This works aims to improve the SoTA of text decoding (language reconstruction) from fMRI with semantics.

Reviewers have raised a number of concerns concerning mainly the evaluation of the performance of the method arguing that, as is, the evidence of improved text decoding with semantic information is too limited.

Given this clear agreement between reviewers (3 out of 4 with strong domain expertise), the paper cannot be endorsed for publication at ICLR.

**Additional Comments On Reviewer Discussion:**

The concerns regarding the metrics and evaluation of the performance of the models are clearly reported by 3 out of 4 reviewers and are legitimate. Discussion between authors and reviewers did not convince.

---

### Decision · Program_Chairs · 2025-01-22

Reject